# Boosting the Adversarial Robustness of Graph Neural Networks: An OOD Perspective

Kuan Li[1*]    Yiwen Chen[2*]    Yang Liu[3]    Jin Wang[4]    Qing He[3]
Minhao Cheng[5†]    Xiang Ao[3†]
[1]The Hong Kong University of Science and Technology    [2]Beihang University
[3]Institute of Computing Technology, Chinese Academy of Sciences    [4]UCLA
[5] Penn State University
```
likuan.ppd@gmail.com
20373456@buaa.edu.cn
{liuyang2023, heqing, aoxiang}@ict.ac.cn
jinwang@cs.ucla.edu
mmc7149@psu.edu
```

## Abstract

Current defenses against graph attacks often rely on certain properties to eliminate structural perturbations by identifying adversarial edges from normal edges. However, this dependence makes defenses vulnerable to adaptive (white-box) attacks from adversaries with the same knowledge. Adversarial training seems to be a feasible way to enhance robustness without reliance on artificially designed properties. However, in this paper, we show that it can lead to models learning incorrect information. To solve this issue, we re-examine graph attacks from the out-of-distribution (OOD) perspective for poisoning and evasion attacks and introduce a novel adversarial training paradigm incorporating OOD detection. This approach strengthens the robustness of Graph Neural Networks (GNNs) without reliance on prior knowledge. To further evaluate adaptive robustness, we develop adaptive attacks against our methods, revealing a trade-off between graph attack efficacy and defensibility. Through extensive experiments over 25,000 perturbed graphs, our method could still maintain good robustness against both adaptive and non-adaptive attacks. The code is provided at https://github.com/likuanppd/GOOD-AT

## 1  Introduction

Adaptive attacks are viewed as a de facto criterion for evaluating the adversarial robustness in vision community (Tramer et al., 2020). Specifically, with full knowledge of the defense model, many defenses can be circumvented by some corresponding schemes (Carlini & Wagner, 2017; Athalye et al., 2018). Recent works show that robust graph neural networks can also be easily bypassed by adaptive attacks (Mujkanovic et al., 2022). Mujkanovic et al. (2022) divided the defense methods into three main categories and seven subcategories and proposed an adaptive attack to one defense per subcategory. The results demonstrate that none of the defenses are as robust as initially assessed in their papers. Adversarial modifications on graphs often violate some intrinsic properties shared by the real-world graphs (Jin et al., 2020a) (e.g., increasing heterophily (Zügner & Günnemann, 2019) and focusing on the high-frequency component (Chang et al., 2021)). The adversary can easily defeat the defenses by imposing constraints on the same properties during the attack (Mujkanovic et al., 2022). Therefore, the key to enhancing adaptive robustness is not relying on artificially defined properties.

Adversarial training appears to be a promising approach, as it does not rely on any prior knowledge of the adversary. Nonetheless, previous studies have demonstrated that adversarial training may not effectively enhance the robustness of GNNs (Xu et al., 2019; Mujkanovic et al., 2022). Our theoretical analysis reveals that adversarial training for graph structures can result in the model learning erroneous (feature, structure)-label mapping. To address this issue, we propose a novel

---
[*]Equal Contribution
[†]Corresponding Authors

adversarial training paradigm that leverages adversarial samples (edges) to help us detect and then remove perturbations, instead of training the model with adversarial training. We argue that the edges generated by attacks are inherently out-of-distribution compared to the original edges. Artificially defined properties used to remove potential perturbation edges reflect incomplete modeling of the out-of-distribution problem with certain inductive biases. To enhance adaptive robustness, we directly use adversarial edges to model the entire OOD problem. Specifically, we generate adversarial edges to train an ensembled OOD detector and then use it to revise the graph during inference. In addition to evasion attacks, we also revisit poisoning attacks through the lens of OOD perspectives.

To further investigate the adaptive robustness, we conduct adaptive attacks against our defenses. We find that these adaptive attacks not only reduce the robustness of our method but also decrease the effectiveness of the attacks themselves. For example, the adaptive design will make the attack ineffective on the vanilla GCN (Kipf & Welling, 2017), a common GNN without any defense mechanism. Consequently, a pivotal insight of our work is that attackers seeking to bolster the adaptivity of their attacks against an OOD-oriented defense may decrease the distribution shift, which, in turn, naturally diminishes the attacks' effectiveness. We identify this phenomenon as the trade-off between attack effectiveness and defensibility.

**Main Contributions**   (1) Based on the evaluation of typical adversarial training, we employ a novel paradigm that leverages the adversarial samples to enhance robustness. (2) Through the lens of OOD, we re-examine graph attacks and defenses and, for the first time, propose the existence of a trade-off between the effectiveness and defensibility of attacks in the context of graph adversarial attacks. (3) We conduct extensive experiments to compare our methods with other baselines in adaptive and non-adaptive settings. Our methods consistently outperform the baselines and SOTA.

## 2   RELATED WORK

Numerous attempts have been made to enhance the robustness of GNNs, and a common approach adopted by most defenses is to leverage the inherent properties of the original graph that are violated by graph adversarial attacks, in order to distinguish between clean and adversarial edges. For example, GNNGuard (Zhang & Zitnik, 2020) and Jaccard-GCN (Wu et al., 2019) find adversarial edges tend to link to nodes in different classes with dissimilar features, so they prune the edges based on this. STABLE (Li et al., 2022) utilizes a similar idea of calculating similarity based on unsupervised representations instead of features. GCN-SVD (Entezari et al., 2020) discovers that attacks exhibit a specific behavior in the spectrum of the graph: high-rank (low-valued) singular components of the graph are affected more than low-rank parts and filter out the high-rank components. GCN-LFR (Chang et al., 2021) studies this problem from the spectral perspective and finds that the perturbations on the low-frequency components are not always smaller than those in the high-frequency ones. ProGNN (Jin et al., 2020b), GSML (Wan & Kokel, 2021), and GCN-GT (Yang et al., 2019) leverage some graph characteristics, e.g., sparsity, low-rankness, and feature smoothness, and incorporates them as regularization terms in the process of structural learning (Zhu et al., 2021).

However, if a defense is designed based on a specific property, it might be vulnerable to adaptive attacks. With full knowledge of the defense model, the adversary can probe for obvious weaknesses or add the differentiable part to the gradient computation during training the attack model (Mujkanovic et al., 2022). These fixed properties can be seen as a specific characterization of OOD. Instead of using crafted fixed properties, we improve the robustness of GNNs by directly reducing the OOD caused by attacks.

## 3   PRELIMINARY

**Notations.**   Let $\mathcal{G} = \{\mathcal{V}, \mathcal{E}\}$ represent an undirected and unweighted graph comprising $N$ nodes. Here, $\mathcal{V}$ and $\mathcal{E}$ (excluding self-loops) are the sets of nodes and edges, respectively. The graph's topology can be represented as a symmetric adjacency matrix $\mathbf{A} \in \{0, 1\}^{N \times N}$, where $\mathbf{A}_{ij} = 1$ indicates that node $v_i$ connects to node $v_j$, and $\mathbf{A}_{ij} = 0$ represents no connection. The original features of all nodes are summarized as a matrix $\mathbf{X} \in \mathbb{R}^{N \times d}$, and $\boldsymbol{x}_i$ indicates the feature of node $v_i$. We use $\mathcal{N}_i$ to denote the first-order neighborhood of node $v_i$, including the node itself. Additionally,

the labels of all nodes are denoted as $\boldsymbol{y}$. Each node has a label $y_i \in \mathcal{C}$, where $\mathcal{C} = c_1, c_2, ..., c_K$. We use $f_\theta(\mathbf{A}, \mathbf{X})$ to represent a GNN, with $\theta$ referring to its parameters.

**Graph Adversarial Attacks**   In this paper, we mainly adopt the most commonly used setting, which considers the global attack on transductive node classification task, but we also conduct experiments in the inductive setting. The attacker's objective is to find an optimal perturbed graph $\hat{\mathcal{G}}$ that maximally impairs the overall performance of the downstream classifier. This can be formulated as follows (Zügner & Günnemann, 2019; Geisler et al., 2021):

$$\operatorname*{argmin}_{\hat{\mathbf{A}} \in \Phi(\mathbf{A})} \quad \mathcal{L}_{atk}(f_{\theta^*}(\hat{\mathbf{A}}, \mathbf{X}), \boldsymbol{y}), \tag{1}$$

where $\hat{\mathbf{A}}$ is the perturbed adjacency matrix, and $\Phi(\mathbf{A})$ is a set of adjacency matrices that satisfies the unnoticeability: $\frac{\|\hat{\mathbf{A}} - \mathbf{A}\|_0}{\|\mathbf{A}\|_0} \leq \Delta$, in which $\Delta$ is the maximum perturbation rate. The $\mathcal{L}_{atk}$ can be computed by the training nodes with ground-truth labels and also by the labels (or pseudo-labels) of the testing nodes. The $\theta^*$ refers to the parameters of the surrogate GNN, which is used to generate perturbations.

**Gradient-based Attacks**   Most attack methods are gradient-based (Chen et al., 2020; Wu et al., 2019; Zügner & Günnemann, 2019; Xu et al., 2019; Geisler et al., 2021), which treat the adjacency matrix as an optimizable parameter and perform malicious modifications on the graph structure by leveraging the gradient of the adjacency matrix with respect to the attack loss. As Eq. (1) is a non-convex discrete optimization problem, some approximations are required for its solution. One way is based on greedy rules, which selects the entry in the adjacency matrix with the largest gradient for flipping, such as Metattack (Zügner & Günnemann, 2019). Another commonly used trick is gradient projection (Xu et al., 2019), which first treats the adjacency matrix as continuous weights, then iteratively updates it using gradients and re-projects it back into the range of $[0, 1]$ to obtain a probability flipping matrix. Finally, the perturbed graph is obtained by sampling according to the probability matrix.

**Poisoning and Evasion**   Poisoning attacks perturb the graph during the model training phase to fool the classifier. Evasion attacks corrupt the graph during the inference phase, where the model is trained on a clean graph and will be tested on a perturbed graph.

**Adaptive Attack**   Adaptive attacks are a type of white-box attack in which the attacker has complete information, including features, graph structure, labels, and all details of the defender's model. Mujkanovic et al. (2022) categorize defense methods into seven types (Günnemann, 2022) and design adaptive attacks for the most representative method in each category. Previous methods are easily bypassed by these adaptive attacks, because they enhance the robustness of GNNs by depending on some specific properties. For instance, Jaccard-GCN (Wu et al., 2019) assumes that perturbed edges tend to connect dissimilar nodes, and thus filters out all edges in the graph with similarity below a certain threshold. An adversary can render the defense strategy ineffective simply by incorporating the same prior knowledge.

## 4   DEFENSE AGAINST EVASION ATTACK

**Graph Adversarial Training vs. Image Adversarial Training.**   Adversarial training is one way to improve robustness that does not rely on specific properties, and it is proven to be the most effective against adversarial attacks in vision domain (Pang et al., 2020; Maini et al., 2020; Pang et al., 2022). Specifically, adversarial training optimizes a hybrid loss function, which is a combination of a standard classification loss and an adversarial loss term:

$$\mathcal{L} = \mathcal{L}_{CELoss}(f(x; \theta), y), \quad \mathcal{L}_{adv} = \max_{x' \in \mathcal{B}(x)} \mathcal{L}(f(x'; \theta), y), \tag{2}$$

where $\mathcal{B}(x) = \{x + \delta \mid \|\delta\|_\infty \leq \epsilon\}$ is the allowed perturbation set. The idea of adversarial training is straightforward: it augments training data with adversarial examples in each training loop, and then the learned model tends to be local invariant for any input: for $\forall x' \in \mathcal{B}(x)$, the model is encouraged to output $y$.

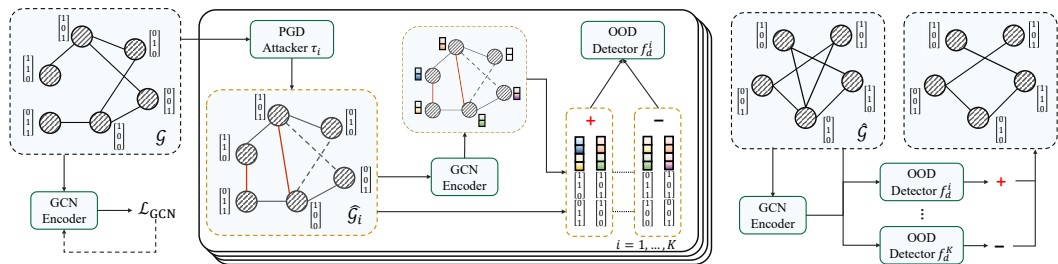

Figure 1: The framework of GOOD-AT. First, we train a GCN on the clean graph, and then we use PGD to generate $K$ perturbed graphs. On each perturbed graph, we label the adversarial edges as positive examples (OOD samples) and the original edges in the graph as negative examples (In-distribution samples). All these samples are used to train a detector. During inference, we leverage an ensemble detector to optimize the graphs structure before using the GCN to predict.

However, it is observed that current graph adversarial training methods for structural attacks do not achieve comparable robustness(Xu et al., 2019; Mujkanovic et al., 2022; Zhang et al., 2022). We theoretically show it will lead to the model learning incorrect mapping relationships.

**Motivation: Failure of Traditional Adversarial training 1.** *Traditional adversarial training for graph structure will result in GNNs learning incorrect (feature, structure)-label mapping. During the adversarial training process, the ground truth label of some nodes has changed, i.e., $\arg\max_y p_d(y \mid x, \mathcal{S}_x) \neq \arg\max_y p_d(y \mid x, \hat{\mathcal{S}}_x)$, but the model is still encouraged to output the label before the change, namely $\arg\max_y p_\theta(y|x, \hat{\mathcal{S}}_x) = \arg\max_y p_d(y \mid x, \mathcal{S}_x)$, where $p_d$ is the ground truth distribution and $p_\theta$ is the model distribution. $\mathcal{S}$ and $\hat{\mathcal{S}}_x$ respectively represent the original and perturbed local structure of node $x$.*

We provide the more details on this motivation in Appendix B, and it demonstrates that directly applying adversarial training from image to graph structure is not feasible. But here's the problem, is it really necessary to apply adversarial training to GNNs? For image data, learning an invariant model via adversarial training is the preferred approach due to the continuous and indistinguishable nature of perturbations, which blend into the original image as indistinguishable noise. However, perturbations on graphs are discrete and separated from clean edges so that they can be directly removed once get identified. With the advantage of this, we turn to utilize these adversarial samples (edges) by training a detector that aims to identify perturbations without depending on any specific properties.

Assuming that the edges on the clean graph are sampled from a ground-truth distribution, the edges generated by the attack algorithm can be viewed as OOD samples on the edge level. From this perspective, previous methods that use artificially defined properties to detect perturbations can be viewed as modeling such OOD detection in an incomplete manner. We avoid this flaw by more comprehensively modeling this problem using neural networks and adversarial samples.

**OOD-detection-based Adversarial Training.** We thus propose our approach Graph OOD-Detection-based Adversarial Training as GOOD-AT. On a high level, GOOD-AT generates perturbations through adversarial attacks during training and uses these adversarial edges as OOD samples, while keeping the initial edges as in-distribution samples. Once we have defined the positive and negative samples, we proceed to train a classifier, namely an OOD detector, on these samples. We train multiple detectors and then combine them into an ensemble detector.

First, we define an OOD detector as a classifier capable of detecting edges that originate from a data-generating distribution (perturbations) distinct from that of the in-distribution edges (clean data). Specifically, the objective of the OOD detection is to identify a suitable decision function (associated with the detector $f_d$) such that for any given input:

$$\Gamma(\boldsymbol{e}; f_d) = \begin{cases} 0, & \boldsymbol{e} \text{ is an in-distribution edge,} \\ 1, & \boldsymbol{e} \text{ is an out-of-distribution edge,} \end{cases} \tag{3}$$

where $e$ is the embedding of an arbitrary edge. In this work, $f_d$ is a binary MLP, and $\Gamma$ is a step function that determines whether an edge is an OOD sample or not based on the output of the detector and a threshold $t$.

Then, we train a two-layer Graph Convolutional Network (GCN) (Kipf & Welling, 2017) $f_{GCN}$ on the clean graph and fix its parameters during the whole training and testing phases. For an detector $f_d^i$, we attack the clean graph by the PGD (Xu et al., 2019) attack $\tau_i$ with tanh logit margin loss (Geisler et al., 2021) to obtain the perturbed graph $\hat{\mathcal{G}}_i$, and the edge representation is computed as followed:

$$e_{uv} = \text{CONCAT} \left[ f_{GCN}(\hat{\mathbf{A}}, \mathbf{X})_u, f_{GCN}(\hat{\mathbf{A}}, \mathbf{X})_v, \boldsymbol{x}_u, \boldsymbol{x}_v \right]. \tag{4}$$

The representation of an edge is obtained by concatenating the representations and the original features of two end nodes $u$ and $v$. To prevent learned representations from becoming unreliable due to significant changes in the local structure of nodes, we add a residual connection scheme, where node features are concatenated at the end. We employ a two-layer MLP as the detector and subsequently train it using the in-distribution samples (original edges) and OOD samples (adversarial edges).

We utilize PGD (Xu et al., 2019) to generate OOD samples by flipping the adjacency matrix in the following way:

$$\hat{\mathbf{A}} = \mathbf{A} + \mathbf{C} \circ \mathbf{S}, \mathbf{C} = \overline{\mathbf{A}} - \mathbf{A}, \tag{5}$$

where $\overline{\mathbf{A}} = 11^T - \mathbf{I} - \mathbf{A}$, and the edge connecting nodes $u$ and $v$ is modified (added or removed) if $\mathbf{S}_{ij} = \mathbf{S}_{ji} = 1$. During gradient-based optimization in PGD, the discrete adjacency matrix is relaxed from $(0, 1)^{N \times N}$ to $[0, 1]^{N \times N}$, and the final weights of the adjacency matrix indicate the probabilities of flipping. The flip matrix $S$ is then sampled from the weighted adjacency matrix. The sampling strategy of PGD guarantees that even with the same parameters, the generated adversarial graphs are distinct each time. To comprehensively model OOD samples, we need to repeat the detector training process multiple times on different perturbed graphs to obtain multiple different detectors. Finally, we have an ensemble detector $f_D = \{f_d^1, f_d^2, ... f_d^K\}$, where $K$ is a hyper-parameter representing the number of detectors.

Training an OOD detector without any OOD data can be challenging, so there is a setting in OOD detection that uses extra OOD training data (OOD Exposure) (Hendrycks et al., 2018; Liu et al., 2020; Bitterwolf et al., 2022; Liu et al., 2023). OOD exposure can significantly improve the performance of OOD detection (Wu et al., 2023; Hendrycks et al., 2018; Liu et al., 2024; Huang et al., 2022), but obtaining OOD data is challenging in real-world scenarios. In graph adversarial attack, we can generate a large number of OOD samples by repeatedly attacking clean graphs, which enables our OOD detector to achieve stronger performance.

**Inference.** During testing, we first use the ensemble detector $f_D$ to determine whether each edge in the graph is an OOD edge:

$$\Delta\left(\boldsymbol{e}; f_D, \Gamma\right) = \begin{cases} \boldsymbol{e} \text{ is an in-distribution edge,} & \text{if } \Gamma(\boldsymbol{e}; f_d^1) = \Gamma(\boldsymbol{e}; f_d^2) = ...\Gamma(\boldsymbol{e}; f_d^K) = 0, \\ \boldsymbol{e} \text{ is an OOD edge,} & \text{if } \exists_{i=1}^K \Gamma(\boldsymbol{e}; f_d^i) = 1. \end{cases} \tag{6}$$

If one detector is confident that this edge is an OOD sample, then it is classified as an adversarial edge. We remove all the detected OOD edges and finally test the classifier $f_{GCN}$ on the revised graph. We provide an overall algorithm of GOOD-AT in Appendix C.

## 5 DEFENSE AGAINST POISONING ATTACK

Previous work has suggested that poisoning attacks can be directly neutralized by certain tricks (Li et al., 2023; Zhan & Pei, 2022), so defense against such attacks is not the focus of this paper. Therefore, we refer to the self-training strategy proposed in (Li et al., 2023) and make minor modifications. This section primarily discusses the working mechanism of poisoning attacks from the out-of-distribution perspective.

In transductive setting, there are two ways in which poisoning attacks can be effective. One is to attack the local structure of training nodes, causing the model to fit incorrect data. The other is to

attack the local structure of testing nodes, causing the model to infer perturbed data. Both types of attacks can occur simultaneously. Attackers can shift the training data away from the original distribution or shift the testing data away as well, and then OOD generalization happens. This is a global-level OOD, which refers to the overall distribution shift between the training set and the testing set. According to Li et al. (2023); Zhan & Pei (2022), the perturbations generated by gradient-based attacks (Zügner & Günnemann, 2019; Xu et al., 2019; Geisler et al., 2021) are nearly all around the training nodes. In the common data split ( 10%/10%/80% (train/val/test)) in graph adversarial attack, the size of training set is relatively small, so concentrating the limited attack budget on the smaller one of training set and testing set can cause a larger distribution shift.

**Self-training Defense.** Knowing that most perturbations are around training nodes, it is straightforward to design a defense strategy - Using the testing nodes' local structure and pseudo-labels for training, i.e., self-training. In poisoning attacks, the defender does not have the clean graph to train an accurate GNN. Therefore, we employ an MLP to generate pseudo-labels instead. Firstly, we train the MLP by labeled samples and then pseudo-label $m$ testing nodes with the highest confidence in each class. Secondly, we isolate the testing graph from the whole graph by pruning the connections between training nodes and testing nodes. Indeed, this defense strategy can be bypassed by some simple adaptive designs, such as imposing the constraint that the generated adversarial edges must be between testing nodes. We will delve into this matter extensively in the following section.

# 6 AN ADAPTIVE VIEW TOWARDS OUR DEFENSES

To more comprehensively assess the adaptive robustness of our proposed technique, we design corresponding adaptive attacks.

**Evasion Attack.** We consider two types of adaptive attacks against GOOD-AT, both assuming that the attacker has access to the detectors. The first approach is to only generate perturbations that cannot be identified by the detectors. In the second approach we incorporate the detector's output as a regularization term in the PGD training loss. However, we find that both strategies will lead to a notable decrease in attack effectiveness. Even the adaptive design will make the attack ineffective on the vanilla GCN (Kipf & Welling, 2017), a normal GNN without any defense mechanism. For the experimental details, see Appendix D).

**Poisoning Attack.** Self-training can be easily circumvented by a strategy that distributes the perturbations across the entire graph. However, if an attacker were to do so, the effectiveness of the attack would be greatly reduced(for details, see Appendix E).

**Trade-off Between Effectiveness And Defensibility.** During the process of the aforementioned adaptive attacks, we observe a trade-off between the defensibility and effectiveness of attack methods: The adaptive designs concurrently mitigate the robustness of our defenses and diminish the inherent efficacy of the attack itself. We postulate that the cause of this phenomenon is that the greater the difference between the generated perturbations and the original edges, the stronger its destructive power, yet the more detectable and defensible. Conversely, the smaller the difference, the less likely it is to cause significant performance degradation, making these adversarial edges less distinguishable from the original graph. Based on this, we propose Hypothesis 1 and elaborate on this point in Appendix F.

**Hypothesis 1.** When facing an ideal OOD detector, the more effective the attack algorithm, the further the perturbations it generates deviate from the ground-truth distribution, which, in turn, makes it easier to be defended against. On the other hand, to increase stealthiness, the effectiveness of the attack method will be reduced.

# 7 EXPERIMENTS

In this section, we empirically analyze the robustness of our methods on both transductive and inductive settings.

## 7.1 SETUP

**Implementation Details**  Following a recent work (Mujkanovic et al., 2022), we compare our methods with the 7 most representative defense methods, i.e., RGCN (Zhu et al., 2019), Jaccard-GCN (Wu et al., 2019), GNNGuard (Zhang & Zitnik, 2020), ProGNN (Jin et al., 2020b), SVD-GCN (Entezari et al., 2020), GRAND (Feng et al., 2020), Soft-Median-GDC (Geisler et al., 2021), on two widely used datasets, namely Cora (Bojchevski & Günnemann, 2017) and Citeseer (Giles et al., 1998). We take two attack methods, PGD (Xu et al., 2019) and Metattack (Zügner & Günnemann, 2019), and a unit test with 7 adaptive attacks into account Mujkanovic et al. (2022). More implementation details can be seen in Appendix A. We only conduct poisoning attacks on ProGNN, as its structure learning is coupled with model training. If we were to perform structure learning in evasion attack, the model would have to be retrained, turning evasion attack into poisoning attack. Therefore, we do not show the performance of ProGNN in evasion attacks.

**Adversarial Unit Test.**  To comprehensively evaluate the robustness of defense methods, Mujkanovic et al. (2022) propose a unit test consisting of seven adaptive attack methods. Referring to (Günnemann, 2022), they categorize the attack methods into seven classes and select the most representative methods from each class as the targets of adaptive attacks. The adversarial graphs generated from these attacks can be encapsulated together to test other defenses, and it can be regarded as the minimum criterion for assessing the adaptive robustness of the defense models. The adaptive attack to the vanilla GCN can be viewed as a non-adaptive attack because GCN is one of the most basic GNNs. In the unit test, the attack budget ranges from 0 to 15%, and each unit test consists of poisoning and evasion attacks, 5 random data splits, with a total of approximately 2700 graphs.

**Metric of Unit Test.**  We adopt the RAUC, a budget-agnostic metric, proposed in Mujkanovic et al. (2022) as the main metric. It means the Relative Area Under the Envelope Curve (AUC), which is the area enclosed by the envelope curve and the accuracy of MLP – a model that is oblivious to the graph structure.

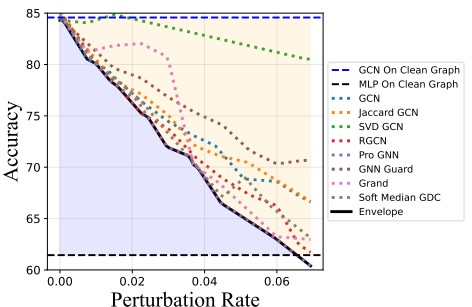

Figure 2: Each dotted line represents an adaptive attack against GCN on the Cora dataset. The best-performing attack at each budget is connected to form the solid black envelope. The blue dashed line represents the clean accuracy of GCN, and the black line represents the accuracy of MLP. The region between the envelope and the accuracy of MLP corresponds to the RAUC. The region between the two dashed lines can be considered the upper bound of RAUC.

For each budget, we can obtain an envelope by selecting the adversarial accuracy of the strongest attack among all attacks. Figure 2 shows an example of the envelope line drawn using GCN.

Formally speaking, $\text{RAUC}(c) = \int_0^{b_0} (c(b) - a_{\text{MLP}})\, \mathrm{d}b$ s.t. $b \lesseqgtr b_0 \Longrightarrow c(b) \gtreqless a_{\text{MLP}}$, where $c(\cdot)$ is a piecewise linear curve representing the robustness per budget, and $a_{\text{MLP}}$ is the accuracy of the MLP. We normalize RAUC so that the RAUC of MLP is 0, and 1 is the maximum value (accuracy remains 100%). RAUC provides a comprehensive evaluation of the robustness of defense methods under various attacks, reflecting the worst-case robustness. The schematic of RAUC is illustrated in Figure 2.

**An Upper Bound of Unit Test.**  We represent the accuracy of GCN on clean data using a straight line and calculate the area enclosed by this line and the accuracy line of MLP. The area can be viewed as an upper bound of RAUC that indicates the model's accuracy does not decrease as the perturbation rates increase. The value of this upper bound is $\text{RAUC}_{max} = 0.61$ on Cora and $\text{RAUC}_{max} = 0.22$ on Citeseer.

## 7.2 UNIT TEST

In this subsection, we present the main experimental results, comparing the adaptive and non-adaptive robustness of our method with other competitive methods using the adversarial unit test.

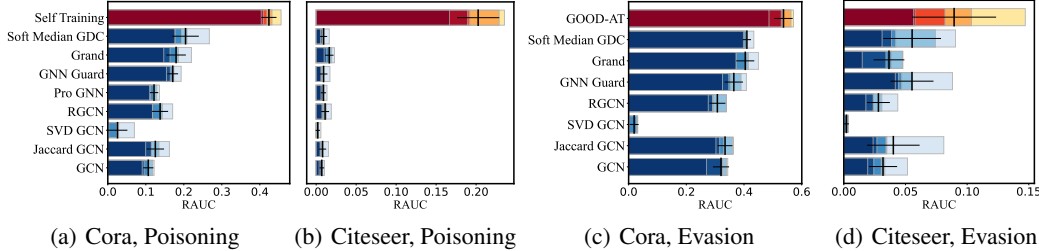

| (a) Cora, Poisoning | (b) Citeseer, Poisoning | (c) Cora, Evasion | (d) Citeseer, Evasion |

Figure 3: RAUC of different defenses on Cora and Citeseer. Each bar plot has five colors representing the results of five different data splits, and the best performance is highlighted by a different color.

**Are Our Methods More Robust Than The Competitive Defenses?** In Fig. 3, we compare the RAUC values of our methods with seven robust GNNs and the vanilla GCN in the adversarial unit test. The black vertical lines represent the mean value of each method, and the horizontal lines represent the standard deviation.

The self-training strategy exhibits extremely significant robustness on poisoning attacks, with RAUC values almost 3-5 times higher than other methods. Especially on Cora, the RAUC of Self-training is close to the upper bound $\text{RAUC}_{max}$. This indicates that existing poisoning attacks must incorporate adaptive attacks against the self-training strategy. We provide a corresponding analysis for this point in Appendix E.

In the evasion attack, GOOD-AT consistently outperformed other methods. The results show that the ensemble detector can indeed help to identify the perturbations. Especially in Cora the RAUC of GOOD-AT is already close to the upper bound $\text{RAUC}_{max}$, indicating that the accuracy of GOOD-AT decreases very slowly as the perturbation rate increases. In addition, the standard deviation of RAUC is relatively small, meaning that this GOOD-AT is not sensitive to data split and remains stable under various partitioning schemes.

The full experimental results of the unit test are provided in Appendix G.2.

**Transferability Study.** Here we conduct a systematic investigation of the transferability of adaptive attacks across different defenses. In Fig. 4, we present the RAUC of each defense model under each adaptive attack and highlight the most robust model for each adaptive attack with a black box. Our method achieves the best performance on all attacks except for the adaptive attack on GCN-SVD. This is because GCN-SVD has weak inherent transferability and relatively poor attack effectiveness on all defenses.

## 7.3 ADDITIONAL ANALYSIS

**Number of Detectors.** Next, we evaluate the impact of the number of the detectors $K$. Table. 1 presents the accuracy of GOOD-AT against PGD on Cora and Citeseer. We observe that the accuracy significantly improves as the number of detectors increases, up to less than 10 detectors. However, the increase in accuracy becomes limited when using 15 detectors. Our other experimental results during the tuning of hyper-parameters indicate that increasing the number of detectors beyond 20 does not yield further improvements.

Table 1: The mean accuracy of GOOD-AT against PGD on Cora and Citeseer with different number of detectors. We highlight the best model with purple.

| Dataset | $K$/Ptb Rate | 0% | 3% | 6% | 9% | 12% | 15% |
|---|---|---|---|---|---|---|---|
| Cora | 5 | 84.51 | 80.94 | 78.47 | 76.61 | 74.6 | 73.39 |
| | 10 | 84.83 | 81.79 | 79.02 | 79.12 | 76.41 | 75.40 |
| | 15 | 84.87 | 82.75 | 80.63 | 79.48 | 77.72 | 76.36 |
| | 20 | 85.31 | 83.16 | 82.93 | 81.14 | 79.28 | 76.81 |
| Citeseer | 5 | 74.76 | 71.50 | 68.48 | 65.71 | 64.87 | 63.51 |
| | 10 | 74.99 | 72.32 | 68.54 | 66.65 | 68.69 | 63.86 |
| | 15 | 74.82 | 72.39 | 69.61 | 69.57 | 68.25 | 67.54 |
| | 20 | 75.54 | 71.75 | 70.68 | 69.49 | 69.19 | 67.82 |

**Inductive Classification.** To further investigate the robustness of GOOD-AT, we evaluate the robustness of GOOD-AT on inductive classification in Appendix G.1 to ensure the model does not

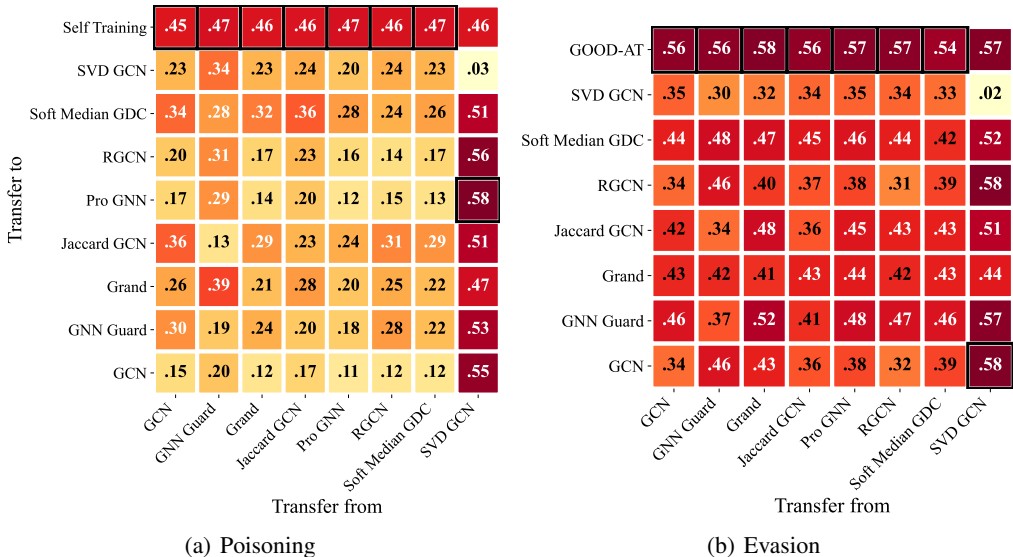

(a) Poisoning (b) Evasion

Figure 4: RAUC values for transferring adaptive attacks designed for one defense to other defenses. Each column represents one adaptive attack, while each row corresponds to a defense model. For the results on Citeseer, see Appendix G.3

utilize the clean local structure of testing nodes. In this setting, during the training process, we remove the testing nodes from the graph to guarantee that clean edges of the testing nodes are not exposed to the classifier and detectors. GOOD-AT outperforms other defenses and likewise demonstrates remarkable robustness.

**Generality.** GOOD-AT is a plug-in method, which can be applied to any GNNs, so we test its generality by changing the base classifier in Appendix G.4. The results demonstrate that substituting GCN with other GNN models not only does not result in a decline in performance but can actually improve robustness. This observation may be attributed to the comparatively weaker representation learning capability of GCN as a basic GNN model.

# 8 LIMITATIONS

While we do not rigorously prove Hypothesis 1 theoretically, we believe it is a valuable concept for research in graph adversarial attacks. This trade-off is primarily based on empirical and intuitive observations from a large number of attack and defense experiments. Notably, this phenomenon is unique to graph attacks. While we leave the more thorough study of this aspect for future research, our findings provide valuable insights into this unique nature of graph attacks and potential avenues for further exploration. We do not consider feature perturbations as most work in this area (Entezari et al., 2020; Geisler et al., 2020; 2021; Lei et al., 2022; Zhang & Zitnik, 2020; Li et al., 2022), and this does not affect our core contributions.

# 9 CONCLUSION

In this paper, we adopt an OOD perspective to re-examine graph adversarial attacks and analyze the distributional shift phenomena in both poisoning and evasion attacks. Our theoretical analysis reveals the limitations of traditional adversarial training in enhancing the adversarial robustness of graph neural networks, leading us to propose a novel adversarial training method that trains multiple OOD detectors to improve the GNN's robustness. Through extensive experiments, we validate the adaptive and non-adaptive robustness of our approach, and our results demonstrate superior performance compared to other methods in both evasion and poisoning attacks.

## 10 ACKNOWLEDGE

The research work is supported by National Key R&D Plan No.2022YFC3303302, the National Natural Science Foundation of China under Grant (No.61976204), and the CAAI Huawei MindSpore Open Fund. Xiang Ao is also supported by the Project of Youth Innovation Promotion Association CAS and the Beijing Nova Program. Yang Liu is also supported by the China Postdoctoral Science Foundation (No. 2023M743567).

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

# A  IMPLEMENTATION DETAILS

**Datasets.**  Following a recent work (Mujkanovic et al., 2022), most of our experiments are conducted on the two most widely used datasets, Cora (Bojchevski & Günnemann, 2017) and Citeseer (Giles et al., 1998). In order to systematically evaluate the robustness of our method and other defenses, we have to relinquish the use of large-scale datasets, as most attacks and defenses are not scalable to such datasets. The data split follows 10%/10%/80% (train/validation/testing). Following other works in graph adversarial attack (Zügner & Günnemann, 2019; Jin et al., 2020a; 2021; Liu et al., 2021), we only use the largest connected component (LCC) of the graphs. The statistics are listed in Table. 2.

Table 2: Dataset statistics.

| Datasets | $N_{LCC}$ | $E_{LCC}$ | Classes | Features |
|---|---|---|---|---|
| Cora | 2,485 | 5,069 | 7 | 1,433 |
| Citeseer | 2,110 | 3,668 | 6 | 3,703 |

**Baselines.**  We evaluated the robustness of 8 graph neural networks under three attack algorithms, Metattack, PGD, and MetaPGD (used only for adaptive unit testing). A brief introduction to these methods is provided below.

- **GCN** (Kipf & Welling, 2017): GCN is a popular graph convolutional network based on spectral theory.

- **RGCN** (Zhu et al., 2019): RGCN leverages Gaussian distributions to represent nodes and employs a variance-based attention mechanism to mitigate the propagation of adversarial attacks.

- **Jaccard-GCN** (Wu et al., 2019): Jaccard-GCN preprocesses the adjacency matrix by removing edges connecting nodes with low Jaccard similarity.

- **GNNGuard** (Zhang & Zitnik, 2020): GNNGuard utilizes cosine similarity to model the edge weights and then calculates edge pruning probability through a non-linear transformation.

- **ProGNN** (Jin et al., 2020b): ProGNN treats the adjacency matrix as a learnable parameter and trains it by minimizing the classification loss and three regularization terms, i.e., feature smoothness, low-rank and sparsity.

- **SVD-GCN** (Entezari et al., 2020): SVD-GCN observe that most of the perturbations lie in the high-rank components of the adjacency matrix, and thus performs low-rank approximation on the adjacency matrix.

- **GRAND** (Feng et al., 2020): GRAND randomly mask the features and adopt a mixed-order propagation, i.e., $\overline{X} = \overline{A}\tilde{X}$, where $\overline{A} = \sum_{k=0}^{K} \frac{1}{K+1}\hat{A}^k$ is a combination of multi-order message passing.

- **Soft-Median-GDC** (Geisler et al., 2021): Soft-Median-GDC first applies GCN normalization and adds self-loops to the adjacency matrix. Then, it preprocesses the matrix with Personalized Page Rank and calculates the weights for message passing based on the distance between neighboring nodes and the median of their neighborhood representations.

- **Metattack** (Zügner & Günnemann, 2019): Metattack uses meta-gradients to solve the bilevel problem underlying poisoning attacks, essentially treating the graph as a hyperparameter to optimize.

- **PGD** (Xu et al., 2019): During gradient-based optimization in PGD, the discrete adjacency matrix is relaxed from $(0,1)^{N \times N}$ to $[0,1]^{N \times N}$, and the final weights of the adjacency matrix indicate the probabilities of flipping.

- **Meta-PGD** (Mujkanovic et al., 2022): Meta-PGD is a combination of PGD and meta gradient. It unrolls the training procedure to obtain gradients in PGD training.

**Hyper-parameters.** We use DeepRobust, an adversarial attack repository (Li et al., 2020), to implement Metattack (Zügner & Günnemann, 2019), PGD (Xu et al., 2019), RGCN (Zhu et al., 2019), Jaccard-GCN (Wu et al., 2019), SVD-GCN (Entezari et al., 2020), and ProGNN (Jin et al., 2020b). GNNGuard(Zhang & Zitnik, 2020), GRAND (Feng et al., 2020), and Soft-Median-GDC (Geisler et al., 2021) are implemented with the code provided by the authors. Mujkanovic et al. (2022) invest a significant effort in tuning the hyperparameters of these models, obtaining more satisfactory accuracies than those reported in their original papers. Therefore, we adopt their hyperparameter values in our experiments.

We tune all the hyper-parameters of our methods based on the results of the validation set. For GOOD-AT, $K$ is the number of detectors and tuned from $\{5, 10, 15, 20\}$. The budgets of PGD used to generate OOD samples are tuned from $0.1 - 1.0$ We consider grid-search for the dimension of hidden layer of the detectors within $32, 64, 128, 256, 512$ and learning rate within $\{0.1, 0.01, 0.001\}$. The threshold of the step function $\Gamma$ is tuned from $\{0.2, 0.5, 0.6, 0.7, 0.8, 0.9\}$. For the GCN classifier, we follow Mujkanovic et al. (2022) to set the drop-out to $0.9$, hidden size to $64$, and weight decay to $0.001$. For self-training, the only hyper-parameter is the number of pseudo-labels in each class, and we tune it from$\{20 - 100\}$.

**Running Environment.** In this study, all experiments were conducted on a computing cluster equipped with NVIDIA Tesla A100 GPUs. Each GPU has 80 GB of memory and is powered by CUDA 11.8. Most of the experiments can be conducted on a single GPU. The operating system used for the experiments is Ubuntu 20.04 LTS. The deep learning models were implemented using PyTorch framework (version 2.0.0) with Python (version 3.8.8) as the programming language. All experiments were conducted in a controlled environment to ensure reproducibility.

## B ADVERSARIAL TRAINING ON GRAPHS

**Details on Motivation 1.** Here we try explain why adversarial training fails on graphs. (Madry et al., 2017) propose adversarial training to boost the robustness in image classification and define the robust error with the Cross-Entropy loss as follow:

$$\mathbf{R}_{\text{ADV}}(\theta) = \left[ \max_{x' \in \mathcal{B}(x)} \mathbb{E}_{p_d(x,y)} \mathcal{L}_{CE}\left(x', y; f_\theta\right) \right] = \mathbb{E}_{p_d(x)} \max_{x' \in \mathcal{B}(x)} - \left[ \sum_y p_d(y \mid x) \log p_\theta(y \mid x') \right], \tag{7}$$

here, the reason why the expectation operator can be placed outside the max operator is that the samples are independent. Trough minimizing the robust error, we have:

$$\arg \min_\theta \mathbb{E}_{p_d(x)} \max_{x' \in \mathcal{B}(x)} - \left[ \sum_y p_d(y \mid x) \log p_\theta(y \mid x') \right]$$

$$= \arg \min_\theta \mathbb{E}_{p_d(x)} \max_{x' \in \mathcal{B}(x)} - \left[ \sum_y p_d(y \mid x) \left( \log p_\theta(y \mid x') - \log p_d(y \mid x) \right) \right] \tag{8}$$

$$= \arg \min_\theta \mathbb{E}_{p_d(x)} \left[ \max_{x' \in \mathcal{B}(x)} \mathbf{KL}(p_d(y|x) \| p_\theta(y|x')) \right],$$

where $p_d$ is the ground-truth distribution $\mathbf{KL}(P\|Q)$ denotes the KL divergence between distribution $P$ and $Q$. The optimal solution of Eq. (8) will encourage $p_\theta(y|x')$ smooth around the sample $x$ and be locally invariant. As the perturbation generated by the attack algorithm also falls within the range of $\mathcal{B}(x)$, the model may maintain the correct output. Here is an assumption that the ground-truth label of the sample in the perturbation range $\mathcal{B}(x)$ remains unchanged, i.e. argmax $p_d(y \mid x) = $ argmax $p_d(y \mid x')$, meaning that the semantic information is preserved. This assumption is reasonable because the budget is often small so the sample $x$ is well bound to a small range around $x$. Additionally, the perturbed images are usually indistinguishable by the human eyes, so we consider the semantic information of the sample remains unchanged. This assumption ensures that the model will not learn incorrect knowledge like incorrect feature-label mappings and is also well-known as unoticeability in graph adversarial attack domain.

Unlike adversarial training in images, which perturbs features (pixels) during training, the most common adversarial training on graphs perturbs the graph structure:

$$\mathbf{R}^{\mathcal{G}}_{\text{ADV}}(\theta) = \max_{\hat{\mathbf{A}} \in \mathcal{B}(\mathbf{A})} \mathbb{E}_{p_d(\mathcal{G} = \{\mathbf{A}, \mathbf{X}\})} \mathcal{L}_{CE}\left(f_\theta\left(\hat{\mathbf{A}}, \mathbf{X}\right), \boldsymbol{y}\right)$$

$$= \max_{\hat{\mathbf{A}} \in \mathcal{B}(\mathbf{A})} -\mathbb{E}_{p_d(\mathcal{G})}\left[\sum_y p_d(y \mid x, \mathcal{S}_x) \log p_\theta(y \mid x, \hat{\mathcal{S}}_x)\right], \tag{9}$$

where $\mathcal{S}$ and $\hat{\mathcal{S}}$ are the substructure of node $x$ in the clean graph and perturbed graph, respectively. They are necessary for message passing in GNN. Then we minimize the robust error on graphs:

$$\arg\min_\theta \mathbf{R}^{\mathcal{G}}_{\text{ADV}}(\theta) = \max_{\hat{\mathbf{A}} \in \mathcal{B}(\mathbf{A})} -\mathbb{E}_{p_d(\mathcal{G})}\left[\sum_y p_d(y \mid x, \mathcal{S}_x) \log p_\theta(y \mid x, \hat{\mathcal{S}}_x)\right]$$

$$= \arg\min_\theta \max_{\hat{\mathbf{A}} \in \mathcal{B}(\mathbf{A})} -\mathbb{E}_{p_d(\mathcal{G})}\left[\sum_y p_d(y \mid x, \mathcal{S}_x)\left(\log p_\theta(y \mid x, \hat{\mathcal{S}}_x) - \log_{p_d} p_d(y \mid x, \mathcal{S}_x)\right)\right]$$

$$= \arg\min_\theta \max_{\hat{\mathbf{A}} \in \mathcal{B}(\mathbf{A})} \mathbb{E}_{p_d(\mathcal{G})}\mathbf{KL}(p_d(y|x, \mathcal{S}_x)) \| p_\theta(y|x, \hat{\mathcal{S}}_x)). \tag{10}$$

The perturbation is defined at the graph level rather than the node level, so the local structures of nodes are not explicitly constrained, allowing for dramatic change to the local structure of each node and resulting in a pronounced dissimilarity between $\hat{\mathcal{S}}_x$ and $\mathcal{S}_x$. Previous works show that attack methods tended to attack low-degree nodes (Zügner & Günnemann, 2019; Li et al., 2022), causing drastic changes in the local structure $\mathcal{S}$ of these nodes and potentially altering their semantic information (Gosch et al., 2023b). In this manner, while the node labels remain unchanged, their local structural semantic information may undergo significant alterations, leading to the invalidation of the smoothness assumption in images, *i.e.*, argmax $p_d(y \mid x, \mathcal{S}_x) \neq$ argmax $p_d(y \mid x, \hat{\mathcal{S}}_x)$. However, while the semantic information changes, the model is still encouraged to output the label before the structural modification, namely $\arg\max_y p_\theta(y|x, \hat{\mathcal{S}}_x) = \arg\max_y p_d(y \mid x, \mathcal{S}_x)$. Thus, this graph-level adversarial training may lead to the model learning incorrect mapping relationships, mapping nodes with completely altered semantic structural information to the original labels. Based on the above analysis, if we want to conduct adversarial training on the structure, a feasible way is to define the constraints at the node or edge level to ensure that the semantic information of nodes remain unchanged.

A recent work (Gosch et al., 2023a) also notices the problem with the too loose definition of the constraint of the perturbations during graph adversarial training. The key difference between our work and theirs is that they attempt to address this problem using local constraints and a flexible GNN (Chien et al., 2020), while we explore a different paradigm of adversarial training. Current adversarial training for GNNs is derived from the visual AT through a one-to-one correspondence. Rather than directly porting existing image AT methods to graph, we advocate for more domain-specific design for graphs. Specifically, compared with the perturbations on images, perturbations on graphs are discrete so that they can be directly removed once get identified. Motivated by this, we relinquish the conventional notion of adversarial training in pursuit of learning an invariant model and propose the GOOD-AT pipeline as a new method of using adversarial samples to enhance the robustness of GNNs. Based on the experimental results, we propose a trade-off between the effectiveness and defensibility of attacks. Although we do not provide a thorough proof for this hypothesis as we claim in the Limitations, we think this idea can be insightful and helpful to the community.

**Unoticeability.** Based on the above analysis, we believe that the semantic information at the node level is altered, and the perturbations are not unnoticeable. So far, the definition of unnoticeability in graph adversarial attack remains an open issue. Prior researches attempt to define it based on degree distribution (Zügner et al., 2018), overall perturbation rate (Zügner & Günnemann, 2019), and graph homophily (Chen et al., 2022), but all these definitions are mostly at the graph level. Based on the aforementioned analysis, a reasonable definition of unnoticeability should be at the sample level. Some attempts have been made in this regard by Gosch et al. (2023b), who uses Bayes optimal

classifiers to determine whether a node's ground-truth label has changed, but Bayes classifiers are often infeasible in real-world data. How to define unnoticeability well is still not well resolved.

## C  ALGORITHM OF GOOD-AT

We provide the overall algorithm of GOOD-AT in Algorithm 1. After determining the number of detectors $K$, we train each detector separately (Line 2-7). Firstly, we use PGD to generate a perturbed graph (Line 3). Then we define the generated perturbed edges as OOD samples and the original edges as in-distribution samples (Line 5). The detector is trained via the positive and negative samples (Line 6). Finally, we ensemble the outputs of all detectors, and purify the graph during inference (Line 8).

---

**Algorithm 1:** GOOD-AT

---

**Input:** Graph $\mathcal{G} = \{\mathcal{V}, \mathcal{E}\}$, Perturbed Graph $\hat{\mathcal{G}}$, Features $\mathbf{X}$, Labels $\boldsymbol{y}_L$, $K$ Detectors, GCN
        Classifier $f_{GCN}$, $K$ PGD Attackers
**Output:** The predicted labels of $\mathcal{V}_U$

---

1   Train the GCN $f_{GCN}$ on labeled nodes $\mathcal{V}_L$               ▷ Start Training;
2   **for** $i=1, ..., K$ **do**
3       Generate the perturbed graph $\hat{\mathcal{G}}_i$ via the PGD attacker $\tau_i$;
4       Generate the representations of edges using Eq. (4);
5       Label the adversarial edges as OOD samples and the original edges as in-distribution
         samples;
6       Train the detector $f_d^i$ by the edges;
7   **end**
8   Optimize the perturbed graph $\hat{\mathcal{G}}$ using Eq. (6)          ▷ Starting Inference;
9   Predict the labels of $\mathcal{V}_U$;
10   Return predicted labels;

---

## D  ADAPTIVE ATTACK TOWARDS GOOD-AT

We design two variants of PGD to perform adaptive attacks on GOOD-AT. These include resampling detected perturbations and incorporating the detector's output as a regularization term in the PGD training loss. Both of them have access to all the information of the trained detectors.

The process of generating adversarial edges using PGD can be devided into three steps. Firstly, the adjacency matrix is relaxed to continuous values in the range during the gradient-based optimization and the resulting weighted change reflects the probability of flipping an edge (Xu et al., 2019; Mujkanovic et al., 2022). Secondly, after each gradient update, the changes are projected back to ensure they do not exceed the permissible budget. Finally, sample the adversarial edges multiple times based on the optimized flipping matrix, selecting one that achieves the best attack performance.

**Resample.** The most direct adaptive attack against GOOD-AT is that we do not generate the perturbations which can be detected. When designing this adaptive attack, we do not modify the first two steps of PGD. In the final step, if the sampled edge can be detected by the detectors, it is discarded, and the corresponding reversal probability in the flipping matrix is set to zero. Sampling then continues until the budget is reached. This ensures that the final generated adversarial edges will not be detected by the detectors in GOOD-AT.

**Regularization.** This adaptive attack modifies the first step of PGD by adding the detector's output as a regularization in the PGD training loss. It can be formulated as:

$$\mathcal{L}_{all} = \mathcal{L}_{atk} + \lambda \mathcal{L}_{reg}, \text{where } \mathcal{L}_{reg} = \frac{1}{N^2 K} \sum_{k=1}^{K} \sum_{i=1}^{N} \sum_{j=1}^{N} \hat{\mathbf{A}}_{ij} f_d^k(e_{ij}), \tag{11}$$

where $f_d^k$ is the $k$-th detector, and $e_{ij}$ is the representation of the edge $\hat{\mathbf{A}}_{ij}$. Considering every element in $\hat{\mathbf{A}}$ would result in a significant computational cost. Therefore, we apply a sparsity operation here, only calculating the elements in the adjacency matrix greater than a threshold $t$.

The experimental results, shown in Table. 3, display the accuracies of GCN under PGD in the first column and GOOD-AT under PGD in the second column. $PGD_{res}$ and $PGD_{reg}$ are resampling and regularization adaptive strategies, respectively.

Table 3: The adaptive robustness on Cora. PGD-GCN indicates GCN is attacked by normal PGD, PGD-GOOD is GOOD-AT under normal PGD, and $PGD_{res}$ is GOOD-AT under PGD with adaptive design.

| Ptb Rate | 3% | 6% | 9% | 12% | 15% |
|---|---|---|---|---|---|
| **PGD-GCN** | $78.26 \pm 1.56$ | $75.10 \pm 0.71$ | $72.15 \pm 1.45$ | $67.83 \pm 1.48$ | $66.39 \pm 1.28$ |
| **PGD-GOOD** | $84.25 \pm 1.90$ | $83.60 \pm 1.77$ | $82.71 \pm 1.14$ | $82.21 \pm 1.73$ | $81.61 \pm 1.10$ |
| **$PGD_{res}$-GOOD** | $82.59 \pm 1.53$ | $81.06 \pm 1.06$ | $79.38 \pm 1.15$ | $77.46 \pm 1.68$ | $76.54 \pm 1.62$ |
| **$PGD_{reg}$-GOOD** | $82.94 \pm 1.50$ | $81.72 \pm 1.33$ | $80.54 \pm 1.66$ | $79.38 \pm 2.01$ | $78.63 \pm 1.40$ |

After incorporating the adaptive design, the robustness of GOOD-AT decreases but still significantly outperforms GCN under normal PGD. We also compare the performance of GOOD-AT and other methods under their respective adaptive attacks (provided by the unit test (Mujkanovic et al., 2022)), as depicted in Figure 5.It can be observed that the adaptive robustness of GOOD-AT is superior to that of other methods.

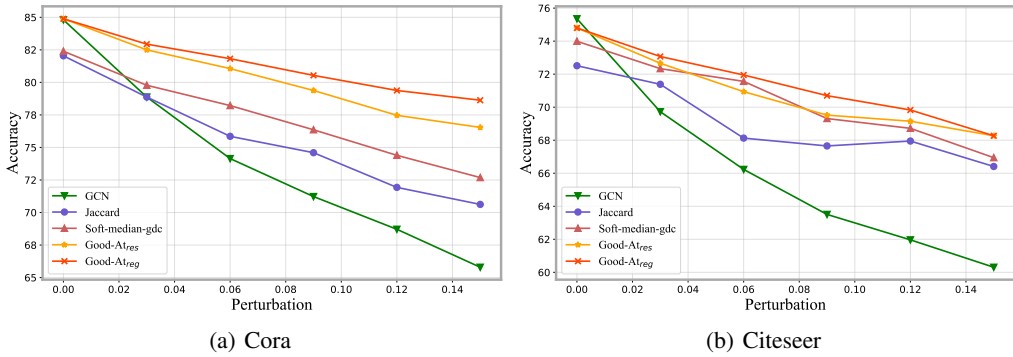

(a) Cora          (b) Citeseer

Figure 5: The mean accuracy of GOOD-AT combined with different GNNs agianst PGD.

It is worth noting that, GOOD-AT is a combination of vanilla GCN and the ensembled detectors. Due to the adversarial perturbations generated by $PGD_{res}$ being able to bypass the detector, GOOD-AT degrades to a vanilla GCN in this scenario. In other words, the transferability of PGD decreases, and the generated perturbations in this case are ineffective against a regular GCN model without any defense. By making the perturbations bypass the detector, the effectiveness of the attack is reduced, which brings a trade-off between effectiveness and defensibility of the attack methods. Hence, perturbations that circumvent detectors are more likely to be in-distribution, which are not that harmful to GNNs.

## E    THE DISTRIBUTION OF PERTURBATIONS IN POISONING ATTACK

**Distribution of Adversarial Edges.** Previous works discover that perturbations generated by the effective poisoning attacks are unevenly distributed on the graph (Li et al., 2023; Zhan & Pei, 2022). Nearly all the adversarial edges are located around the training nodes. Specifically, we can divide the edges into three groups:

- **Group 1**: The edges connect two training nodes.

Table 4: Distribution of adversarial edges on Cora

| Edges | Ptb Rate | Group 1 | Group 2 | Group 3 |
|---|---|---|---|---|
| **Clean** | | 0.7% | 18.3% | 81.0% |
| **Metattack** | 5% | 5.1% | 94.1% | 0.8% |
| | 10% | 11.9% | 87.5% | 0.6% |
| | 15% | 9.1% | 88.7% | 2.2% |
| | 20% | 7.8% | 90.7% | 1.5% |
| **DICE** | 5% | 0.9% | 17.1% | 82.0% |
| | 10% | 0.9% | 16.8% | 83.3% |
| | 15% | 1.0% | 17.0% | 82.0% |
| | 20% | 0.9% | 17.4% | 81.7% |
| **PGD** | 5% | 28.6% | 70.2% | 1.2% |
| | 10% | 24.6% | 73.7% | 1.7% |
| | 15% | 21.7% | 77.0% | 1.3% |
| | 20% | 23.9% | 74.9% | 1.2% |

- **Group 2**: The edges connect a training node and a testing node.

- **Group 3**: The edges connect two testing nodes.

We provide the distribution of the adversarial edges and original edges on Cora in Table. 4. As training nodes account for 80% of all nodes, Group 3 is the largest part of the clean graph. However, Metattack allocates nearly all budgets around the training nodes, i.e., in Group 1 and Group 2. DICE (Waniek et al., 2018) is a heuristic method that directly increases the heterophily of the graph by randomly connecting nodes from different classes and disconnecting nodes from the same class, and we observe that the distribution of perturbation generated by DICE is almost identical to that of the original edges, indicating the existence of bias in gradient-based attacks.

(Li et al., 2023) provide an explanation for this phenomenon. In poisoning attacks, the effectiveness of the attack algorithm comes from increasing the distribution shift between the training set and the testing set, and a smaller training set means that limited perturbations can be more effective in increasing the distance between the training and testing distributions when applied to the training set. This makes poisoning attacks that are effective, with perturbations focused on the training set, very easy to defensd, such as using a self-training strategy or training the model with the validation set (Zhan & Pei, 2022).

The distribution of the attack algorithm in the graph can be adjusted by the mask used during the attack (Li et al., 2023), i.e., selecting which nodes to compute the attack loss $l_{atk}$. For example, there are two commonly used loss in graph attack, $\mathcal{L}_{train} = \mathcal{L}(f_{\theta^*}(\hat{\mathbf{A}}, \mathbf{X})_L, \boldsymbol{y}_L)$ and $\mathcal{L}_{self} = \mathcal{L}(f_{\theta^*}(\hat{\mathbf{A}}, \mathbf{X})_U, \hat{\boldsymbol{y}}_U)$. The former calculates the loss by the predictions of GNN and labels of training nodes, while the latter uses those of testing nodes. In $\mathcal{L}_{self}$, $\hat{\boldsymbol{y}}_U$ are the pseudo-labels in gray-box attack and ground-truth labels in white-box attack. $L_{train}$ is calculated based on the local structure of the training nodes, and therefore, during backpropagation, gradients are only propagated to the portion of the adjacency matrix corresponding to the training nodes. Similarly, attacks with $L_{self}$ can only affect the local structure of testing nodes.

**Adaptive Attack.** From the perspective of a white-box attack, we can bypass the self-training defense by spreading perturbations throughout the entire graph.

Distributing the attack across the entire graph may effectively bypass self-training defense, but it could also significantly reduce the effectiveness of the attack. We adapt Metattack to uniformly distribute the perturbations and improve DICE by fixing its attack to the training nodes. Table. 5 shows the results of these two attacks on GCN. The efficacy of Metattack significantly declines, whereas DICE shows notable improvement when the perturbations are concentrated on training nodes, demonstrating that distributing the attack across the entire graph can potentially compromise

the performance. If attackers opt to target test nodes due to self-training strategy, they inherently compromise their attack performance.

Table 5: The performance of Metattack and DICE on Cora. The asterisk means the modification of the basic models.

| Ptb Rate | Meta | Meta* | DICE | DICE* |
|:---:|:---:|:---:|:---:|:---:|
| 5% | 76.36 | 79.23 (↗2.87) | 81.89 | 80.68 (↘1.21) |
| 10% | 71.62 | 75.60 (↗5.98) | 80.92 | 78.12 (↘2.80) |
| 15% | 66.37 | 71.37 (↗5.00) | 79.93 | 75.35 (↘4.58) |
| 20% | 60.31 | 66.84 (↗6.53) | 77.10 | 73.79 (↘3.31) |

## F  TRADE-OFF BETWEEN EFFECTIVENESS AND DEFENSIBILITY

We observe from the experimental results in Appendix E and Appendix D that once the attacker attempts to consider circumventing our defenses, the effectiveness of the attack will significantly decrease. This is not exhibited by adaptive attacks designed for defense methods. According to the results in Fig. 4, the adaptive attacks for other defenses have strong transferability.

Referring to Theorem 4.1 in Li et al. (2023), the efficacy of graph attacks stems from their capacity to increase the distribution shift. The greater the difference between the perturbations and the original distribution, whether in terms of global graph distribution or local edge distribution, the more destructive the perturbations are to GNNs. However, from the perspective of OOD detection, samples that deviate further from the original distribution are more easily detected. In order to enhance the stealthiness of an attack, the attacker must sacrifice a portion of the efficacy of the perturbations. For example, in poisoning attack, if attackers opt to spread perturbations throughout the whole graph due to self-training strategy, the global distribution shift (between training and testing nodes) caused by the attack will not be as significant as when the attack is concentrated solely on the training nodes, so they inherently compromise their attack performance. In evasion attack, to evade detection by a ideal detector as GOOD-AT, the attacker must relinquish the generation of those OOD adversarial edges and instead select edges that are relatively in distribution. Additionally, a neural network based detector possibly considers all the potential properties that can be used to distinguish between adversarial and normal edges. When an attacker incorporates such a detector as a regularization term into their attack loss, the generated edges become similar to normal edges. Obviously, these edges inflict less damage on the performance of the GNNs.

Achieving complete robustness against adversarial attack is not realistic with existing technologies, but we speculate that there may exist a balance point in graph structural attack, where neither the attacker nor the defender can further enhance their performance by improving their strategies. A detector that meets this condition can be referred to as an ideal detector. At this stage, relying more on specific inductive biases can improve its robustness against certain type of attacks, but incorporating the corresponding adaptive design into the attack will render the defense ineffective. In this work, we do not provide a thorough or theoretical proof for this point, so it is more of a conjecture. We leave the exploration of this balance point for future research.

## G  MORE EXPERIMENTS

### G.1  INDUCTIVE CLASSIFICATION

Under the transductive setting, evasion defense assumes knowing the clean graph, including test nodes. To further investigate the robustness of GOOD-AT, we conduct experiments within an inductive setting to ensure that clean edges of the testing nodes are not exposed to the classifier and detectors. We randomly selected 10% of the nodes for training, 80% for validation, and the remaining 10% for testing. During the training process, we removed the testing nodes from the graph.

- Training: During the training process, the testing nodes and the edges connected to them are removed so that we only use the subgraph of training and validation nodes to train a GCN

and detectors. This ensures that clean testing data are not used. To generate adversarial edges, utilized for training the detectors, we use PGD to attack validation nodes. Throughout these operations, a GCN model served as the surrogate model.

- Testing: During the testing phase, we employ PGD on the entire graph targeting the testing nodes initially, followed by utilizing a detector to cleanse the graph. Finally, we use the trained GCN to predict the labels of testing nodes.

This scenario is similar to node injection, where we ensure that clean structural information of testing nodes is not used during training. The experimental results on Cora are listed in Table. 6. GOOD-AT outperforms other defenses and still exhibits strong robustness. This indicates that GOOD-AT does not rely on utilizing the clean structure of testing nodes during training. Even in the scenario where the features and local structures of testing nodes are completely unknown, the detectors in GOOD-AT are capable of detecting adversarial edges.

Table 6: The inductive performance on Cora. We highlight the best model with purple and the runner-up with brown.

| Dataset | Ptb Rate | GCN | GOOD-AT | Soft-Median | Jaccard | SVD-GCN |
|---------|----------|-----|---------|-------------|---------|---------|
| Cora | 2.5% | $75.40 \pm 2.11$ | $\mathbf{79.21 \pm 2.34}$ | $\mathbf{75.89 \pm 2.13}$ | $75.03 \pm 1.86$ | $74.82 \pm 2.12$ |
| | 5% | $68.72 \pm 2.37$ | $\mathbf{78.58 \pm 2.58}$ | $72.36 \pm 2.73$ | $\mathbf{73.18 \pm 3.06}$ | $68.06 \pm 3.07$ |
| | 7.5% | $63.10 \pm 2.36$ | $\mathbf{76.91 \pm 1.99}$ | $\mathbf{70.65 \pm 2.82}$ | $70.60 \pm 2.41$ | $62.51 \pm 3.69$ |
| | 10% | $59.28 \pm 1.73$ | $\mathbf{76.00 \pm 2.23}$ | $67.22 \pm 2.75$ | $\mathbf{70.33 \pm 3.27}$ | $58.90 \pm 2.81$ |
| Citeseer | 2.5% | $68.25 \pm 2.41$ | $\mathbf{71.62 \pm 2.07}$ | $\mathbf{69.80 \pm 1.56}$ | $69.07 \pm 1.96$ | $67.83 \pm 2.63$ |
| | 5% | $61.83 \pm 2.62$ | $\mathbf{70.54 \pm 2.58}$ | $\mathbf{66.38 \pm 2.30}$ | $65.42 \pm 1.02$ | $61.55 \pm 2.41$ |
| | 7.5% | $55.51 \pm 2.00$ | $\mathbf{69.88 \pm 2.16}$ | $\mathbf{65.11 \pm 0.99}$ | $63.34 \pm 3.13$ | $55.49 \pm 2.85$ |
| | 10% | $49.76 \pm 1.91$ | $\mathbf{69.22 \pm 2.79}$ | $\mathbf{62.30 \pm 2.57}$ | $59.81 \pm 1.88$ | $50.03 \pm 3.37$ |

## G.2 THE FULL RESULTS OF THE UNIT TEST

There are 5 random data split in the unit test, and we report the full results on the split-0.

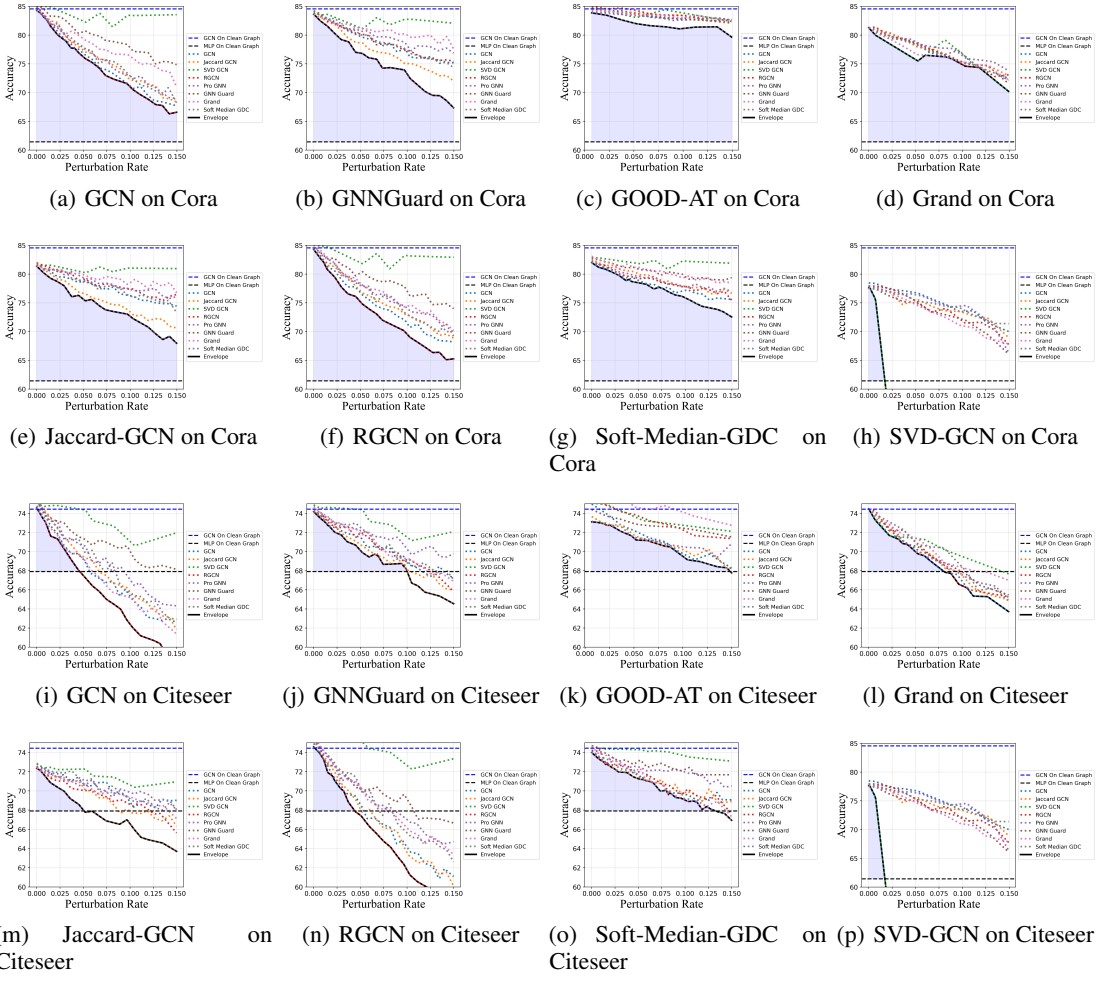

Figure 6: Unit test of evasion attack

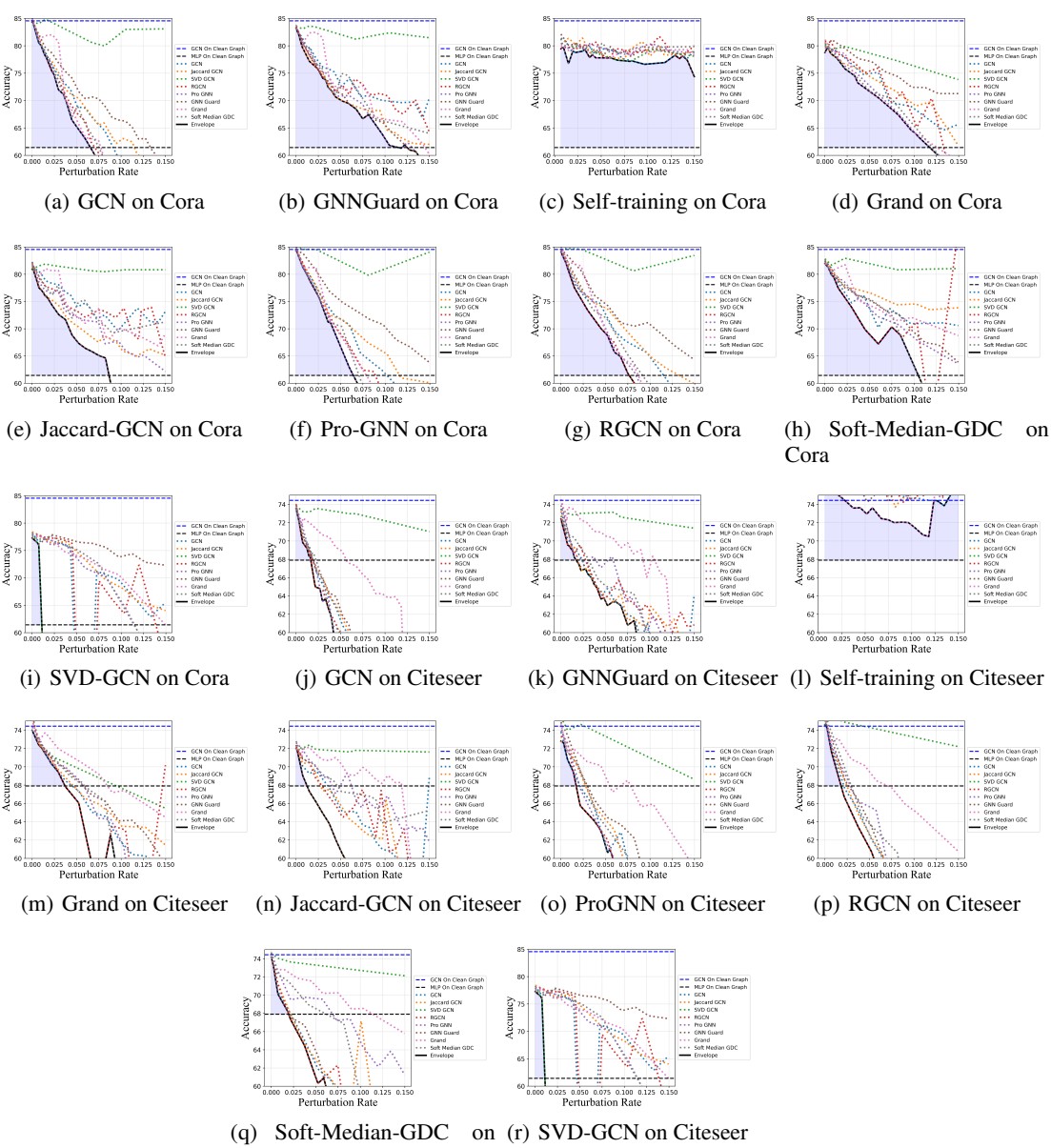

Figure 7: Unit test of poisoning attack

## G.3 TRANSFER ATTACK ON CITESEER

The results on Citeseer are consistent with those on Cora is shown in Fig. 8, but Citeseer is noticeably much more vulnerable when facing attacks. This conclusion aligns with the findings of Mujkanovic et al (Mujkanovic et al., 2022).

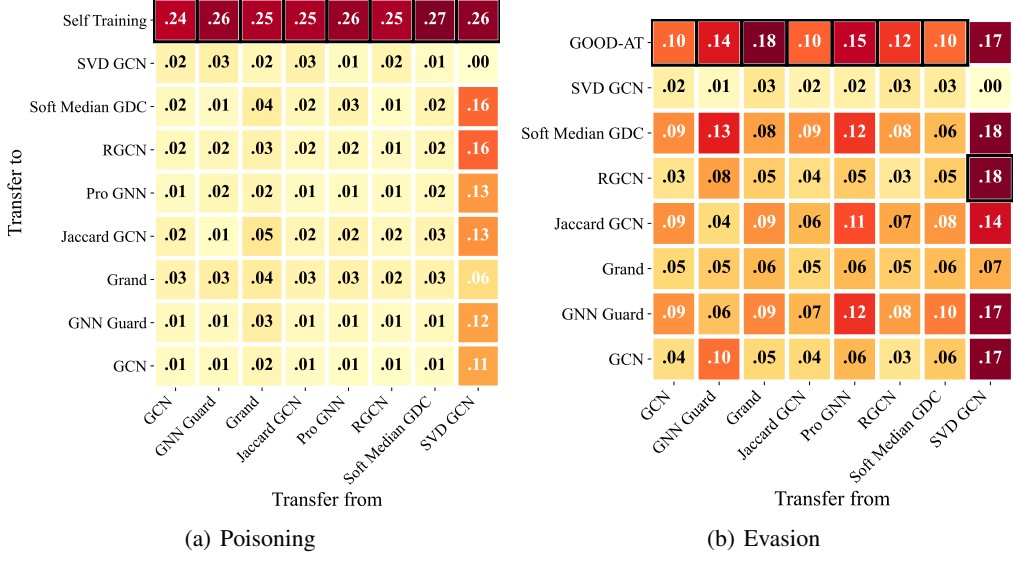

(a) Poisoning

(b) Evasion

Figure 8: RAUC values for transferring adaptive attacks designed for one defense to other defenses. Each column represents one adaptive attack, while each row corresponds to a defense model.

## G.4 GENERALITY OF GOOD-AT

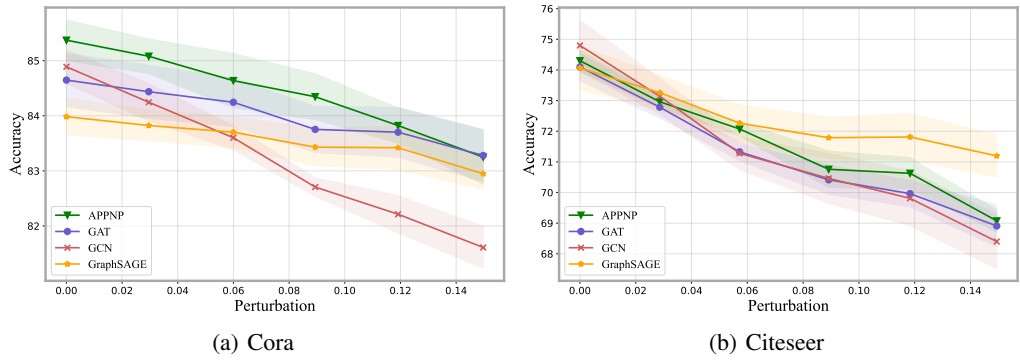

(a) Cora

(b) Citeseer

Figure 9: The accuracy of GOOD-AT using different GNNs as base classifier against PGD.

GOOD-AT is a plug-in training approach that can be combined with any GNN. To verify its generality, we replaced the base classifier with GraphSAGE (Hamilton et al., 2017), GAT (Veličković et al., 2017), and APPNP (Gasteiger et al., 2018). The results, as shown in Fig. 9, demonstrate that replacing GCN with other GNN models does not lead to performance degradation. On the contrary, GCN exhibits the lowest overall robustness. We speculate that the reason for this could be that GCN, as one of the earliest and fundamental GNN models, may have a relatively weaker representation learning capability compared to other GNNs. From this experiment, we can conclude that GOOD-AT demonstrates strong generality and can be transferred to any GNN model.

The OOD detectors are trained specifically for an individual GNN classifier instance, raising a question that can it transfer well to a different GNN instance. To verify this, we incorporate the OOD

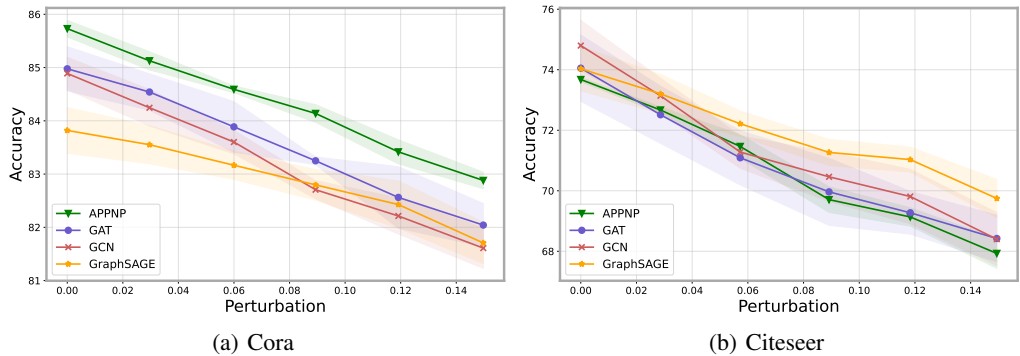

(a) Cora  (b) Citeseer

Figure 10: The accuracy of GOOD-AT using different downstream GNN classifiers against PGD.

detector which is trained by GCN with other downstream GNN models, including GAT, GraphSAGE, and APPNP. The results in Fig. 10 demonstrate that the ensemble detector still can successfully defend against attack.

## H  AGAINST OTHER ATTACKS

GOOD-AT is trained on the adversarial edges generated by PGD, raising a question that can it successfully defend against other attacks. To address this concern, we test GOOD-AT on two other attacks PR-BCD and GR-BCD (Geisler et al., 2021). From the Table. 7, we can observe that attacking with PR-BCD and GR-BCD on PGD-trained GOOD-AT are also not effective. Compared to image perturbation, structural perturbations have a much smaller search space due to its discreteness. Therefore, we surmise that the OOD edges generated by different attack methods are similar within this limited range.

Table 7: The performance of GOOD-AT against PR-BCD and GR-BCD on Cora

| Ptb Rate | $N_{LCC}$ | $E_{LCC}$ | Classes |
|---|---|---|---|
| 3% | 84.25 | 84.17 | 84.36 |
| 6% | 83.60 | 83.85 | 84.06 |
| 9% | 82.71 | 82.44 | 83.73 |
| 12% | 82.21 | 82.78 | 83.54 |
| 15% | 81.61 | 82.03 | 83.22 |

