# OpenReview forum: "Boosting the Adversarial Robustness of Graph Neural Networks: An OOD Perspective"
_ICLR.cc/2024/Conference — ICLR 2024 poster_

### Official Review · Reviewer_TfP9 · 2023-10-27

**Soundness:** 3 good
**Presentation:** 3 good
**Contribution:** 3 good
**Rating:** 8
**Confidence:** 4

**Summary:**

The authors present a method to improve GNN adversarial robustness. The approach trains $K$ OOD edge detection MLPs that classify edges based on their internal representations of a GCN model and their input attributes. At inference time, the OOD detector ensemble predicts for each edge whether it is potentially adversarial or not, and removes the edge according to that decision. The authors present evidence based on a GNN robustness benchmark suggesting their defense outperforms existing ones. Further, the authors study robustness to poisoning attacks as well as inductive evasion attacks.

**Strengths:**

* The authors present compelling results on a diverse robustness benchmark
* The approach is clear and well-motivated
* The authors consider a wide range of settings including transductive evasion and poisoning attacks as well as inductive attacks

**Weaknesses:**

* While the range of settings considered is wide, the set of models and dataset considered is quite narrow in return
* The transductive evasion defense could potentially be reduced to the trivial perfect defense mentioned in [Gosch et al. 2023]
* Regarding Proposition 1, the "proof" isn't really a proof, and, does not show a problem with adversarial training for graphs, but highlights that the definition of the perturbation set is too loose.

**Questions:**

* My most pressing concern is regarding the very recent work [Gosch et al. 2023]. I acknowledge that it is so recent that it is not reasonable to expect the authors to have it in their paper, yet their results are very relevant for this work. Specifically, the authors propose a trivial perfect defense for evasion attacks in transductive setting, which effectively memorizes the clean input graph and ignores the potentially perturbed graph at inference time (Proposition 1 of the referenced work). Assuming unique node attributes, wouldn't the perfect version of the OOD ensemble defense reduce to this trivial defense?
* On a relate note, [Gosch et al. 2023] also notice the problem with the too loose definition of the set of allowed perturbations, and, in turn, present a method that employs local constraints. How does this affect Proposition 1 in this work?
* How is the threshold $t$ determined? Is it the same across OOD detectors in the ensemble?
* Typically, ensembles work via majority vote. In this work, the authors flag an edge as potentially adversarial if **one** of the detectors in the ensemble does. What is the reason for this?
* The OOD detectors are trained specifically for an individual GNN classifier instance. I wonder if it is also possible to transfer the OOD detector ensemble to a different GNN instance?

References
---
Gosch, L., Geisler, S., Sturm, D., Charpentier, B., Zügner, D., & Günnemann, S. (2023). Adversarial Training for Graph Neural Networks. NeurIPS 2023. https://arxiv.org/abs/2306.15427

---

> ### Author Response · Authors · 2023-11-11
> **The first reply to reviewer TfP9**
>
> We would like to thank the reviewer for the detailed and insightful feedback! Please see below for our response to the concerns:
>
> >*Compare with [1]*
>
> [1] and we both notice the problem with the too loose definition of the constraint of the perturbations during graph adversarial training, and we specify this point by the mathematical formulation of supervised learning and adversarial training. The key difference between our work and theirs is that they attempt to address this problem using local constraints (we also suggest this in the Appendix B), while we explore a different paradigm of adversarial training. Current adversarial training for GNNs is derived from the visual AT through a one-to-one correspondence. Rather than directly porting existing visual AT methods to graph, we advocate for more domain-specific design for graphs. Specifically, compared with the perturbations on images, perturbations on graphs are discrete so that they can be directly removed once get identified. Motivated by this, we relinquish the conventional notion of adversarial training in pursuit of learning an invariant model and propose the GOOD-AT pipeline as a new method of using adversarial samples to enhance the robustness of GNNs. Based on the experimental results, we propose a trade-off between the effectiveness and defensibility of attacks. Although we do not provide a thorough proof for this hypothesis as we claim in the Limitations, we think this idea can be insightful and helpful to the community.
>
> After the proceeding of NeurIPS, we found this paper too, but we have already submitted this draft. We add the discussion with [1] in the rebuttal revision (red font).
>
> >*The perfect defense for evasion attacks in transductive setting*
>
> While achieving perfect performance through this method is possible, we consider that transductive node classification is a cornerstone of GNN research, and studying its robustness remains valuable and important. Additionally, to further investigate the robustness of GOOD-AT and avoid using the clean structure of testing nodes, we conduct inductive experiments in the Appendix G.1. The experimental results demonstrate that GOOD-AT also works well in this scenario.
>
> >*How is the threshold t determined? Is it the same across OOD detectors in the ensemble?*
>
> Yes, it is the same for all the detectors, and we search for the best threshold from 0.1-0.5.
>
> >*Typically, ensembles work via majority vote. In this work, the authors flag an edge as potentially adversarial if one of the detectors in the ensemble does. What is the reason for this?*
>
> Robustness considers worst-case generalization, so we believe that the strictest perturbation detection mechanism should be employed. If even one detector identifies an edge as a potential perturbation, it should be removed rather than voted upon. Intuitively, if a small number of detectors identify an edge as a potential perturbation, it should be removed for safety reasons. It is better to err on the side of caution.
>
> >*The OOD detectors are trained specifically for an individual GNN classifier instance. I wonder if it is also possible to transfer the OOD detector ensemble to a different GNN instance?*
>
> This is a great point, and we add some experiments in the Appendix G.4. We transfer the OOD detector to various GNNs, including GAT, GraphSAGE, and APPNP. The results in Fig. 10 demonstrate that the ensemble detector still can successfully defend against attack.
>
> >*While the range of settings considered is wide, the set of models and dataset considered is quite narrow in return*
>
> Our choice of settings and datasets follows the first research work on graph adaptive robustness [2]. Larger datasets are barely possible in the unit test since most defenses are not very scalable. Below, we present the experimental results on relatively larger graph Pubmed against PR-BCD. We can observe that GOOD-AT still exhibits strong robustness.
>
> |Ptb rate|GCN|GOOD-AT|
> |-|-|-|
> |5%|83.25|83.35|
> |10%|80.92|83.31|
> |15%|78.99|83.31|
> |20%|77.12|83.18|
>
> >*Proof of proposition 1*
>
> In adversarial training for images, it is usually assumed that the semantic information of the sample within the perturbation budget remains unchanged. However, in graphs, this loose constraint leads to changes in the semantic information of the nodes, while the node labels remain unchanged. This can cause the model to learn an erroneous mapping between semantic information and labels. We formulate a proposition and provide a proof to articulate this viewpoint, as we attempted to explain this observation through the mathematical formulation of supervised learning and adversarial training. We will consider to change the word “proof” by words like “analysis”.
>
>
>
> ## References
> [1] Gosch, L., Geisler, S., Sturm, D., Charpentier, B., Zügner, D., & Günnemann, S. (2023). Adversarial Training for Graph Neural Networks. In NeurIPS, 2023.
>
> [2] Are Defenses for Graph Neural Networks Robust? In NeurIPS, 2022.

---

> > ### Comment · Reviewer_TfP9 · 2023-11-21
> >
> > Dear authors,
> >
> > Thank you for thoroughly addressing my comments and concerns. I have increased my score accordingly.

---

> > > ### Author Response · Authors · 2023-11-21
> > > **Thank you for the response**
> > >
> > > Thank you for your recognition of our work! We would like to discuss this paper if you have any remaining concerns before the end of the discussion period.

---

> ### Author Response · Authors · 2023-11-19
> **Looking forward to your reply**
>
> Dear Reviewer TfP9, since the discussion period is  is ending in less than three days, we would greatly appreciate it if you could take a look at our reply to your review and let us know if you have any remaining questions. We look forward to addressing any remaining concerns before the end of the discussion period.

---

### Official Review · Reviewer_sgRe · 2023-11-01

**Soundness:** 2 fair
**Presentation:** 2 fair
**Contribution:** 2 fair
**Rating:** 3
**Confidence:** 5

**Summary:**

The paper adopts an out-of-distribution perspective to re-examine graph adversarial attacks and analyze the distributional shift phenomena in both poisoning and evasion attacks at graph and edge levels. The authors propose an adversarial training method that trains multiple OOD detectors to improve the GNN’s robustness. Through extensive experiments, we validate the adaptive and non-adaptive robustness of our approach.

**Strengths:**

1. The authors show that  the simple adversarial training will lead to the model learning incorrect knowledge
2. The authors conduct extensive experiments over 25,000 graphs to compare the robustness of our methods with other baselines

**Weaknesses:**

1. The paper is hard to follow
2. The authors show that adversarial edges are OOD, which is straightforward
3. Notations are hard to understand.
4. The tested defenses are already shown to be vulnerable to adaptive attacks
5. Lack of comparison with provable defense results.

**Questions:**

Proposition 1 is very hard to follow

What are the key differences between OOD-detection-based Adversarial Training vs. standard adversarial training?

The tested defenses, shown in Figure 2,  are already shown to be vulnerable to adaptive attacks. Why do you choose them as baselines?

Why not evaluating the results of certified defense? How about the comparison with them?

---

> ### Author Response · Authors · 2023-11-11
> **The first reply to reviewer sgRe**
>
> We thank for the reviewer’s time.
>
> **Before we reply to the questions. After careful consideration, we have decided to express our disappointment upon receiving this review.** It appears that this review has been largely accomplished by replicating the content of our paper. For example, the summary and the strengths are verbatim copies of our sentences in the draft. Even the reviewer forgot to modify the subject, using the first-person plural "we" and "our" in the summary and strengths. We make a comparison between the review and our sentences below and highlight the same parts are in bold.
>
> >*Reviewer’s summary:*
>
> The paper **adopts an out-of-distribution perspective to re-examine graph adversarial attacks and analyze the distributional shift phenomena in both poisoning and evasion attacks at graph and edge levels.** The authors **propose an adversarial training method that trains multiple OOD detectors to improve the GNN’s robustness. Through extensive experiments, we validate the adaptive and non-adaptive robustness of our approach.**
>
> >*Our sentences in the Conclusion Section:*
>
> **we adopt an OOD perspective to re-examine graph adversarial attacks and analyze the distributional shift phenomena in both poisoning and evasion attacks at graph and edge levels.** […] leading us to **propose a novel adversarial training method that trains multiple OOD detectors to improve the GNN’s robustness. Through extensive experiments, we validate the adaptive and non-adaptive robustness of our approach.**
>
> >*Strengths written by reviewer:*
>
> 1. The authors show **that the simple adversarial training will lead to the model learning incorrect knowledge**
>
> 2. The authors **conduct extensive experiments over 25,000 graphs to compare the robustness of our methods with other baselines.**
>
> >*Our sentences in the contributions in the Abstract Section:*
>
> 1.	we theoretically prove **that the simple adversarial training will lead to the model learning incorrect knowledge.**
>
> 2. We **conduct extensive experiments over 25,000 graphs to compare the robustness of our methods with other baselines.**
>
> **The questions raised by the reviewer are mostly common knowledge in this field.** This review may not foster valuable discussion or help us improve our paper. We feel that our work has not been given sufficient attention by the reviewer. We sincerely hope the reviewer can reread our paper according to our answers to the questions.

---

> ### Author Response · Authors · 2023-11-15
> **Answers to the questions and concerns**
>
> ## Questions:
>
> >*What are the key differences between OOD-detection-based Adversarial Training vs. standard adversarial training?*
>
> Standard AT uses the adversarial samples to train the task classifier. In contrast, ours utilizes the adversarial edges and original edges to train a detector, which is used to detect and remove the perturbations during testing.
>
> >*The tested defenses, shown in Figure 2, are already shown to be vulnerable to adaptive attacks. Why do you choose them as baselines?*
>
> [1] is the first research work that investigates the adaptive robustness of GNNs. They demonstrate that current robust GNNs are highly vulnerable to adaptive attacks. As we specify in the paper, we choose the same baselines as [1], and the baselines are selected not because they are vulnerable to adaptive attack, but rather their status as the quintessential representatives of seven categories of defense methods [1]. **Improving adaptive robustness of GNN is an unsolved problem that is the focus of our endeavors in this study.** The core of our work is to develop a defense mechanism with adaptive robustness. We do not cherry-pick any specific robust GNNs, nor do we select them based on their poor performance. **We are trying to figure out why they are weak and solve it.**
>
> >*Hard to follow or understand; Notations are hard to understand*
>
> We employ the most used notations in the GNN community, and we will include a notation table in the appendix. The logical chain of our work is that we first argue that previous methods are not adaptively robust because they rely on specific properties of the graph. They can be easily defeated by adversaries using the same knowledge. Adversarial training is one way without need for any prior knowledge, so it might be a solution. However, previous adversarial training is directly ported from image adversarial training, raising some drawbacks on graphs. In adversarial training for images, it is usually assumed that the semantic information of the sample within the perturbation budget remains unchanged. However, in graphs, this loose constraint leads to changes in the semantic information of the nodes, while the node labels remain unchanged. This can cause the model to learn an erroneous mapping between semantic information and labels (Proposition 1). We advocate for designing some new way of using adversarial samples to enhance adaptive robustness with domain-specific designs. Thus, we propose our new method (GOOD-AT), which is a totally pipeline of leveraging adversarial samples. To analyze the adaptive robustness of our method, we design two adaptive attacks against it. And during the experiments, we find a new phenomenon about the trade-off between the effectiveness and defensibility of attacks (Hypothesis 1).
>
> >*Why not evaluating the results of certified defense? How about the comparison with them?*
>
> Certified robustness is another line of studying robustness, which targets provable robustness which does not lend itself to empirical evaluation. Certificates typically provide a lower bound on the actual robustness while attacks provide an upper bound [8]. Moreover, to the best of our knowledge, none of the highly cited work in this area compare their robustness with certified defenses [1-7].
>
> [1] All You Need Is Low (Rank): Defending Against Adversarial Attacks on Graphs. In WSDM, 2020.
>
> [2] Reliable graph neural networks via robust aggregation. In NeurIPS, 2020.
>
> [3] Gnnguard: Defending graph neural networks against adversarial attacks. In NeurIPS, 2020.
>
> [4] Robustness of graph neural networks at scale. In NeurIPS, 2022.
>
> [5] Graph structure learning for robust graph neural networks. In KDD, 2020.
>
> [6] Topology Attack and Defense for Graph Neural Networks: An Optimization Perspective. In IJCAI, 2019.
>
> [7] Are Defenses for Graph Neural Networks Robust? In NeurIPS, 2022.

---

> > ### Comment · Reviewer_sgRe · 2023-11-22
> > **Reply to Authors' Response**
> >
> > Thanks for the response. However, my main concerns are not well addressed.
> >
> > First, the proposition 1 is based on the assumption that many nodes’ local structure are largely changed.  However, when the attack is pretty strong (i.e., white-box attacks), this would not be satisfied on many nodes. Also, I do not think proposition 1 is a proposition, it is just an observation that is not verified.
> >
> > Second, the authors train a detector that aims to identify perturbations without depending on any specific properties, because they treat adversarial edges as OOD edges for simplicity. However, I really doubt a detector can detect adversarial edges vs in-distribution edges, especially under the strong white-box attack. Imaging that if this is the case, why are the empirical defenses based on adversarial edge detection failed? Adversarial edges are OOD edges for sure, and are far more stealthy than OOD.
> >
> > Third, the previous works did not test the certified defenses because the certified defenses were not proposed before year 2020. There are several works on certified defense against graph evasion/poisoning attacks after year 2021. As it has already been demonstrated that all the empirical defenses are broken, why do not test certified defenses?
> >
> > Overall, I think the proposed empirical defense is just a delta improvement over the existing ones and I am doubtful about its  soundness, especially for the adversarial edge detector.

---

> ### Author Response · Authors · 2023-11-22
> **The second reply to reviewr sgRe**
>
> We thank the reviewer for the reply and let me carefully address the concerns.
>
> >*The change of the local structure in adaptive attacks*
>
> The semantic information of a node's local structure will change basically is **not an assumption**, as this point is thorough discussed in [1]. It is not entirely clear what is meant by 'pretty strong' attacks, but if we are referring to adaptive white-box attacks, [2] provides experimental evidence that many properties of adaptive attacks are not significantly different from those of non-white-box attacks, including node degree, closeness centrality, homophily, jaccard similarity, and the ratio of removed edges. Therefore, it is reasonable to conclude that adaptive and non-adaptive attacks lacking local constraints can result in changes to the semantic information of a node's local structure.
>
> >*I really doubt a detector can detect adversarial edges vs in-distribution edges, especially under the strong white-box attack. Imaging that if this is the case, why are the empirical defenses based on adversarial edge detection failed? Adversarial edges are OOD edges for sure, and are far more stealthy than OOD.*
>
> **Why can detect adversarial edges** - The detectors are trained by adversarial and normal edges as positive and negative samples, respectively, so it is able to discriminate adversarial edges (OOD) and original edges (In-distribution). To further demonstrate that the detectors successfully remove the perturbations, we show the statistics of the removed edges on Cora compared with Jaccard. We can observe that GOOD-AT nearly remove all the perturbations (against PGD and perturbation rate is nearly 9%).
>
> |Total perturbation|GOOD-AT|Jaccard|
> |-|-|-|
> |415|386|217|
>
> **Why other adversarial edge detection failed** - I’m not sure which methods the reviewer is indicating here. If the reviewer could provide some references, we could discuss this more precisely. Previous methods did not directly model the OOD phenomenon, but instead tried to use specifically defined knowledge to filter out perturbed edges, e.g., node similarity [10, 11] and low-rank approximation [12, 13]. Therefore, they are vulnerable to the adversaries that leverage the same knowledge. In contrast, we directly model the OOD problem by utilizing the adversarial edges. The detector is trained using adversarial and normal edges as positive and negative samples, which can be seen as a comprehensive consideration of various properties.
>
> **Why our method is robust under white-box attack** - We provide two kinds of adaptive attacks (white-box attacks) on GOOD-AT that the adversary has access to all the information of the detectors in Section 6 and Appendix D, and the results show that the effectiveness of the attack decreases while bypassing the detector. This is not difficult to understand, because if a perturbed edge tries to maintain consistency with the original edges on various properties (not a single pre-defined property), it will not be similar to normal edges so that it’s not very harmful. This is the trade-off between defensibility and effectiveness that we have discovered. To our knowledge, this is the first defense method that takes into account adaptive robustness.
>
> >*Compare with provable robustness*
>
> The experimental settings and focuses for provable robustness and empirical robustness are entirely different, which is why provable robustness does not lend itself to empirical evaluation [2, 9].
>
> Provable works do not compare with empirical methods: [3-5]
>
> Empirical methods do not compare with provable robustness: [6-8]
>
> All above citations are latest works in this area and have been published after 2022.
>
> >*delta improvement*
>
> Our proposed method is not incrementally based on any existing method but rather introduces a brand-new pipeline that leverages adversarial samples to enhance robustness. As the logical chain is clarified in the first reply to the reviewer, we believe that the reasons behind the adaptive and non-adaptive robustness of our method are quite clear.

---

> > ### Author Response · Authors · 2023-11-22
> > **References of the second reply**
> >
> > [1] Gosch, Lukas, et al. "Revisiting Robustness in Graph Machine Learning." In ICLR 2023.
> >
> > [2] Mujkanovic, Felix, et al. "Are Defenses for Graph Neural Networks Robust?." In NeurIPS 2022.
> >
> > [3] Scholten, Yan, et al. "Randomized message-interception smoothing: Gray-box certificates for graph neural networks." In NeurIPS 2022.
> >
> > [4] Schuchardt, Jan, Yan Scholten, and Stephan Günnemann. "(Provable) Adversarial Robustness for Group Equivariant Tasks: Graphs, Point Clouds, Molecules, and More." In NeurIPS 2023.
> >
> > [5] Schuchardt, Jan, et al. "Localized Randomized Smoothing for Collective Robustness Certification." In ICLR 2023.
> >
> > [6] Li, Kuan, et al. "Revisiting graph adversarial attack and defense from a data distribution perspective." In ICLR 2023.
> >
> > [7] Lei, Runlin, et al. "Evennet: Ignoring odd-hop neighbors improves robustness of graph neural networks."In NeurIPS 2022.
> >
> > [8] Jin, Wei, et al. "Empowering Graph Representation Learning with Test-Time Graph Transformation." In ICLR 2023.
> >
> > [9] Stephan Günnemann. Graph neural networks: Adversarial robustness. In Lingfei Wu, Peng Cui, Jian Pei, and Liang Zhao, editors, Graph Neural Networks: Foundations, Frontiers, and Applications, chapter 8, . Springer, 2021.
> >
> > [10] Wu, Huijun, et al. "Adversarial examples for graph data: deep insights into attack and defense." In IJCAI 2019.
> >
> > [11] Zhang, Xiang, and Marinka Zitnik. "Gnnguard: Defending graph neural networks against adversarial attacks." In NeurIPS 2020.
> >
> > [12] Entezari, Negin, et al. "All you need is low (rank) defending against adversarial attacks on graphs." In WSDM 2020.
> >
> > [13] Chang, Heng, et al. "Not all low-pass filters are robust in graph convolutional networks." In NeurIPS 2021.

---

### Official Review · Reviewer_hRXc · 2023-11-01

**Soundness:** 4 excellent
**Presentation:** 4 excellent
**Contribution:** 3 good
**Rating:** 6
**Confidence:** 4

**Summary:**

In this paper, the authors theoretically show that AT can lead to models learning incorrect information.
To overcome this issue, they use AT paradigm incorporating OOD detection.

**Strengths:**

1) The idea of suggesting AT paradigm with OOD detection on GNN is novel.

2) The theoretical part is sound.

3) Experiments are sound. I specifically liked that they tested their solution even when the attacker has full knowledge of the detectors, and not only the standard model.

4) paper is well written and easy to follow.

**Weaknesses:**

One question I had is - if the detector is an ensemble, how will it work against an EoT adversary? Can the authors test this kind of scenario?

**Questions:**

See Weaknesses section.

---

> ### Author Response · Authors · 2023-11-11
> **The initial reply to reviewer hRXc**
>
> We would like to thank you for your positive feedback! Let me answer your question below.
>
> >*if the detector is an ensemble, how will it work against an EoT adversary? Can the authors test this kind of scenario?*
>
> Typical adversarial attack tries to attack the original inputs, while EoT extends this to maximize the expectation of log-likelihood given transformed inputs. The problem of EoT is widely studied in vision community. Nevertheless, the problem of EoT has not yet been explored in the realm of graph adversarial attack Additionally, graph data and image data exhibit significant distinctions, potentially lacking a physical interpretation. On the other hand, defining the transformation of graph data also presents a challenge.
>
> This is a meaningful direction and should be thoroughly studied, but it is bit out of the scope of this paper. We are happy to work on it in the future.

---

> ### Author Response · Authors · 2023-11-19
> **Looking forward to your reply**
>
> Dear Reviewer hRXc, since the discussion period is is ending in less than three days, we would greatly appreciate it if you could take a look at our reply to your review and let us know if you have any remaining questions. We look forward to addressing any remaining concerns before the end of the discussion period.

---

> > ### Comment · Reviewer_hRXc · 2023-11-22
> > **Replay to authors**
> >
> > Dear authors,
> >
> > Since my question remained for future research, I don't have any further questions at this point.
> >
> > Good luck!

---

> > > ### Author Response · Authors · 2023-11-22
> > > **Thanks for your reply**
> > >
> > > We thank the reviewer for the reply! We are willing to discuss any remaining problems before the end of the discussion period.

---

### Official Review · Reviewer_RUQz · 2023-11-03

**Soundness:** 3 good
**Presentation:** 2 fair
**Contribution:** 2 fair
**Rating:** 6
**Confidence:** 3

**Summary:**

This paper studies improving the robustness of graph neural networks (GNN) to both evasion and poisoning attacks. The main idea is to model the generated adversarial edges from existing attacks as out of distribution (OOD) data, and trains OOD detectors remove adversarially perturbed edges. Empirical results show that the proposed approach outperforms existing baseline defenses in various settings.

**Strengths:**

1. The idea of using the OOD nature of adversarial edges to train OOD detectors and remove perturbed edges is interesting.
2. The proposed defense also achieves remarkable performance against different state-of-the-art attacks compared to existing baseline defenses.

**Weaknesses:**

1. Hypothesis 1 is the key high-level motivation for designing OOD detector based defenses. However, I am not sure how much this hypothesis can help in practical applications. Specifically, it is indeed true that the attacks can be more destructive when the distributions shift more significantly. However, we measure the empirical vulnerability by comparing the accuracy drops before and after the attack, and there might exist a situation where the attack is considerably successful (from practical point of view) without being significantly different from the in-distributions.
2. The OOD samples are generated by PGD attacks, which can be restricted. The current experimental setting does not eliminate the possibility of adaptive attackers targeting the OOD detector, meaning there might exist some OOD samples that are still effective at inducing distribution shifts, but are not similar to PGD induced OOD samples.
3. In the poisoning scenario, the whole process is based on the assumption that effective poisoning perturbations happen in the training nodes. This again does not capture the fact that, if it is simply impossible for attackers to induce more effective poisoning attacks by targeting other nodes in the graph.

**Questions:**

The common theme of the 3 weaknesses above is that the proposed approach does not provide a more fundamental reasoning on the effectiveness of the proposed approach. If the authors have a more fine-grained analysis, the quality of the paper will be improved.

---

> ### Author Response · Authors · 2023-11-11
> **The initial reply to reviewer RUQz**
>
> We would like to thank the reviewer for the constructive comments and questions. Let us answer the questions below.
>
> >*it is indeed true that the attacks can be more destructive when the distributions shift more significantly. However, we measure the empirical vulnerability by comparing the accuracy drops before and after the attack, and there might exist a situation where the attack is considerably successful (from practical point of view) without being significantly different from the in-distributions.*
>
> This is a good point, and we also consider this potential scenario in which the perturbations generated by the attack are relatively in-distribution and then can evade detection, thereby raising doubts about the efficacy of the attack under such conditions. Consequently, in Appendix D, we propose an adaptive attack strategy, premised on the assumption that the attacker has full knowledge of the detector, thereby guaranteeing that the perturbations generated are in-distribution and go undetected. Our experimental results (Table 3.) demonstrate that even when confronting the vanilla GCN, which lack robustness design, the efficacy of the adaptive attack significantly decreases. As a result, the in-distribution attack seems to be much weaker than OOD attacks. On the other hand, **most edges** in the original graph are helpful for the downstream tasks, so it is reasonable to assume that in-distribution edges (**like most edges**) are not that harmful for GNNs. This trade-off is not exhibited by adaptive attacks designed for other defense methods. According to the results in Figure 4, the adaptive attacks for other defenses have strong transferability. **Hence, this hypothesis provides practical instructions for designing future adaptive defenses against structural perturbations, emphasizing the need for comprehensively considering the OOD phenomenon of perturbations, rather than relying on specific properties.** Such a holistic consideration would compel attack methods to be unable to circumvent defenses without compromising their own efficacy.
>
> Additionally, this hypothesis serves as a complementary analysis, without undermining the core contributions --- We scrutinize the reasons for the poor performance of adversarial training on graphs and introduced a novel paradigm for employing adversarial samples. Moreover, we evaluate its adaptive robustness through unit testing and adaptive attack strategies. To the best of our knowledge, this is the first defense method on graphs considering adaptive robustness.
>
> >*The OOD samples are generated by PGD attacks, which can be restricted. The current experimental setting does not eliminate the possibility of adaptive attackers targeting the OOD detector, meaning there might exist some OOD samples that are still effective at inducing distribution shifts, but are not similar to PGD induced OOD samples.*
>
> To address this concern, we conduct the following experiment. We first train the GOOD-AT by PGD attack and then test its efficacy on PR-BCD and GR-BCD attacks [3] on Cora.
>
> |Ptb rate|PGD|PR-BCD|GR-BCD|
> |-|-|-|-|
> |3%|84.25|84.17|84.36|
> |6%|83.60|83.85|84.06|
> |9%|82.71|82.44|83.73|
> |12%|82.21|82.78|83.54|
> |15%|81.61|82.03|83.22|
>
> From the table, we can observe that attacking with PR-BCD and LR-BCD on PGD-trained GOOD-AT are also not effective. Compared to image perturbation, structural perturbations have a much smaller search space due to its discreteness. Therefore, we surmise that the OOD edges generated by different attack methods are similar within this limited range.
>
> >*In the poisoning scenario, the whole process is based on the assumption that effective poisoning perturbations happen in the training nodes. This again does not capture the fact that, if it is simply impossible for attackers to induce more effective poisoning attacks by targeting other nodes in the graph.*
>
> We agree with the reviewer that attacking testing nodes is feasible. Experimental results from [1, 2] demonstrate that poisoning attacks are more effective than evasion attacks, and [2] speculate that the heightened efficacy of poisoning attacks stems from the concentration of perturbations on the training nodes. If attackers opt to target testing nodes (like evasion attack) due to self-training strategy, they inherently compromise their attack performance, highlighting the trade-off we previously mentioned. This already signifies a level of robustness. We elaborate more on this point in the Appendix E. poisoning attack is not the focus of our paper, and here we mainly want to explain it from an OOD perspective and point out that this trade-off also exists in poisoning attack.
>
> **We add the above analysis and experiment in the rebuttal revision (red font).**
>
> [1] Are Defenses for Graph Neural Networks Robust? In NeurIPS, 2022.
>
> [2] REVISITING GRAPH ADVERSARIAL ATTACK AND DEFENSE FROM A DATA DISTRIBUTION PERSPECTIVE. In ICLR, 2023.
>
> [3] Robustness of graph neural networks at scale. In NeurIPS, 2022.

---

> ### Author Response · Authors · 2023-11-19
> **Looking forward to your reply**
>
> Dear Reviewer RUQz, since the discussion period is is ending in less than three days, we would greatly appreciate it if you could take a look at our reply to your review and let us know if you have any remaining questions. We look forward to addressing any remaining concerns before the end of the discussion period.

---

> > ### Comment · Reviewer_RUQz · 2023-11-22
> > **Thanks for the clarifications**
> >
> > Thanks for the quick and detailed response to my concerns. However, they still lack some formal type of guarantees and is unclear if the proposed approach is indeed inherently robust. However, all current evaluations and also results from existing literature well support the main claim in the paper and so, I wouldn't be too harsh on this. I am accordingly increasing my score.

---

> > > ### Author Response · Authors · 2023-11-22
> > > **Thank you for the response**
> > >
> > > Thank you for your recognition of our main claim and contribution! We would like to discuss this paper if you have any remaining concerns before the end of the discussion period.

---

> > > > ### Comment · Reviewer_RUQz · 2023-11-22
> > > >
> > > > To be more precise about what I mean by lacking some formal guarantees, I mean there is no formal guarantee on why such a method can in principle work, as it is based on conjecture or hypothesis. It might still be the case that current robustness are artifacts of limitation in current empirical attacks. I admit this can be very challenging and that is why I said I would not be too harsh on this. I would like to make it clear to the authors I am willing to defend this paper.

---

> ### Author Response · Authors · 2023-11-22
> **Thanks for your clarification**
>
> We basically agree with you. As far as we know, we are the first defense work that considers adaptive robustness, and we also propose two kinds of white-box attacks to test our defense method and conduct adaptive unit tests. This indicates that our method is adaptively robust under current empirical evaluation. However, there may be attacks or new adaptive ideas in the future that can defeat our defense. But at this moment, we believe that our work is insightful and valuable to the graph adversarial attack community. Adversarial training is an interesting problem on graphs that has not been completely solved, and meanwhile it is very diffrent from visual AT, so another reason our work is crucial for the field is that we advocate from start to finish that adversarial training should not simply be transplanted from image to graph, but rather the nature of perturbations (e.g., discreteness) on graphs should be taken into account when designing the pipeline.
>
> **Once again, we sincerely thank you for your serious attitude and recognition of our work. We are greatly encouraged.**

---

### Meta-Review · Area_Chair_diHE · 2023-12-12

**Metareview:**

The paper makes an interesting, novel contribution and is well positioned w.r.t. the state of the art, despite its deficiencies.

Reviewers appreciated the effort made by the authors to address raised concerns, and their engagement during the discussion period.

A recurrent concern is, as put forth by reviewer RUQz, the absence of a formal guarantee, and the potential coupling of the observed performance with the limitations current empirical attacks employed in the paper. The paper does nevertheless make a contribution, both through the novelty of the proposed overall approach via an OOD point of view as well as through an extensive experimentation. The authors are strongly encouraged to include additional experiments and insights from the discussion in their final version of the paper.

Proposition 1 should not be a proposition, and it's "Proof" should *not* be rephrased as "Analysis"; words like "the DNN is encouraged" do not belong in a formal statement. This vague/informal treatment appears to have detracted from the overall quality/clarity of the paper. The authors should change this section to an expository form (it could even by titled "Motivation: Failure of Traditional Adversarial training", describing the phenomenon, giving a high level intuition why it happens, referencing other papers that observe it (including Gosch et al.)  and pointing to the appendix for more details. Again, do not call this "Analysis".

**Justification For Why Not Higher Score:**

The paper tackles a difficult problem and still has some issues, but ...

**Justification For Why Not Lower Score:**

...it would be worth seeing this published and communicated in the community: it has a fresh way of looking at the problem, is timely, and well positioned w.r.t. sota.

---

### Decision · Program_Chairs · 2024-01-16

Accept (poster)